# LEARNING TO EXPLORE FOR STOCHASTIC GRADIENT MCMC

## ABSTRACT

Bayesian Neural Networks (BNNs) with high-dimensional parameters pose a challenge for posterior inference due to the multi-modality of the posterior distributions. Stochastic Gradient Markov Chain Monte Carlo (SGMCMC) with cyclical learning rate scheduling is a promising solution, but it requires a large number of sampling steps to explore high-dimensional multi-modal posteriors, making it computationally expensive. In this paper, we propose a meta-learning strategy to build SGMCMC which can efficiently explore the multi-modal target distributions. Our algorithm allows the learned SGMCMC to quickly explore the high-density region of the posterior landscape. Also, we show that this exploration property is transferrable to various tasks, even for the ones unseen during a meta-training stage. Using popular image classification benchmarks and a variety of downstream tasks, we demonstrate that our method significantly improves the sampling efficiency, achieving better performance than vanilla SGMCMC without incurring significant computational overhead.

## 1 INTRODUCTION

Bayesian methods have received a lot of attention as powerful tools for improving the reliability of machine learning models. Bayesian methods are gaining prominence due to their ability to offer probability distributions over model parameters, thereby enabling the quantification of uncertainty in predictions. They find primary utility in safety-critical domains like autonomous driving, medical diagnosis, and finance, where the accurate modeling of prediction uncertainty often takes precedence over the predictions themselves. The integration of Bayesian modeling with (deep) neural networks, often referred to as Bayesian Neural Networks (BNNs), introduces exciting prospects for the development of secure and trustworthy decision-making systems.

However, there are significant problems for the successful application of BNNs in real-world scenarios. Bayesian inference in high-dimensional parameter space, especially for deep and large models employed for the applications mentioned above, is notoriously computationally expensive and often intractable due to the complexity of the posterior distribution. Moreover, posterior landscapes of BNNs frequently display multi-modality, where multiple high density regions exist, posing a significant challenge to efficient exploration and sampling. Due to this difficulty, the methods that are reported to work well for relatively small models, for instance, variational inference (Blei & McAuliffe, 2017) or Hamiltonian Monte Carlo (HMC) (Neal et al., 2011), can severely fail for deep neural networks trained with large amount of data, when applied without care.

Recently, Stochastic Gradient Markov Chain Monte Carlo (SGMCMC) methods (Welling & Teh, 2011; Chen et al., 2014; Ma et al., 2015) have emerged as powerful tools for enhancing the scalability of approximate Bayesian inference. This advancement has opened up the possibilities of applying Bayesian methods to large-scale machine learning tasks. SGMCMC offers a versatile array of methods for constructing Markov chains that converge towards the target posterior distributions. The simulation of these chains primarily relies on stochastic gradients, making them particularly suitable for BNNs trained on large-scale datasets. However, despite the notable successes of SGMCMC in some BNN applications (Welling & Teh, 2011; Chen et al., 2014; Ma et al., 2015; Zhang et al., 2020), there remains a notable challenge. Achieving optimal performance often demands extensive engineering efforts and hyperparameter tuning. This fine-tuning process typically involves human trial and error or resource-intensive cross-validation procedures. Furthermore, it's worth noting that

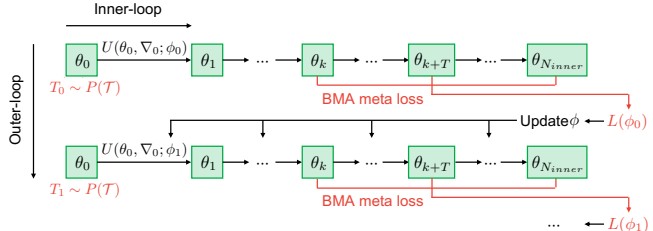

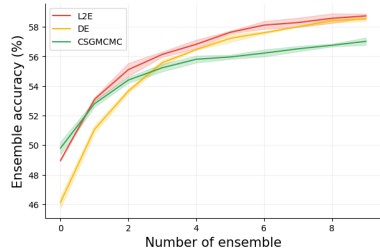

Figure 1: Meta-learning procedure of Learning to Explore (L2E).

Figure 2: Ensemble accuracy for different number of samples on Tiny-ImageNet.

even with the use of SGMCMC methods, there remains room for improvement in efficiently exploring multi-modal posterior distributions. As a result, in practical applications, a trade-off between precision and computational resources often becomes necessary.

To address these challenges, we introduce a novel meta-learning framework tailored to enhance the efficiency of SGMCMC algorithms. Traditional SGMCMC methods often rely on handcrafted design choices inspired by mathematical or physics principles, such as the formulation of kinetic energy terms, curl, and diffusion matrices. Recognizing the pivotal role these design components play in shaping the trade-off between exploration and exploitation within SGMCMC chains, we argue in favor of learning them directly from data rather than manually specifying them. To achieve this, we construct neural networks to serve as meta-models responsible for approximating the gradients of kinetic energy terms. These meta-models are trained using a diverse set of BNNs inference tasks, encompassing various datasets and architectural configurations. Our proposed approach, termed L2E, exhibits several advantageous properties, including better mixing rates, improved prediction performance, and a reduced need for laborious hyperparameter tuning. We point out that ours is not the first meta learning algorithm for SGMCMC, as Gong et al. (2018) already explored a similar concept. However, they did not achieve to learn appropriate exploration-exploitation balance in simulating multi-modal BNNs posterior, and their algorithm was not demonstrated to scale for large-scale BNNs with robust architecture and dataset transfer, limiting the practicality.

Our contributions can be summarized as follows:

- We introduce L2E, a novel meta-learning framework enhancing SGMCMC methods. In contrast to conventional hand-designed approaches and meta-learning approach (Gong et al., 2018), L2E learns the kinetic energy term directly, offering a more data-driven and adaptable solution.

- We present a multitask training pipeline equipped with a scalable gradient estimator for L2E. This framework allows the meta-learned SGMCMC techniques to generalize effectively across a wide range of tasks, extending their applicability beyond the scope of tasks encountered during meta-training.

- Using real-world image classification benchmarks, we demonstrate the remarkable performance of BNNs inferred using the SGMCMC algorithm discovered by L2E, both in terms of prediction accuracy and sampling efficiency.

## 2 BACKGROUNDS

### 2.1 SGMCMC FOR BAYESIAN NEURAL NETWORKS

**Settings.** In this paper, we focus on supervised learning problems with a training dataset $\mathcal{D} = \{(x_i, y_i)_{i=1}^n$ with $x_i$ being observation and $y_i$ being label. Given a neural network with a parameter $\theta \in \mathbb{R}^d$, a likelihood $p(y \mid x, \theta)$ and a prior $p(\theta)$ are set up, together defining an energy function $U(\theta) = -\sum_{i=1}^n \log p(y_i \mid x_i, \theta) - \log p(\theta)$. The goal is to infer the posterior distribution $p(\theta \mid \mathcal{D}) \propto \exp(-U(\theta))$. When the size of the dataset $n$ is large, evaluating the energy function $U(\theta)$ or its gradient $\nabla_\theta U(\theta)$ may be undesirably costly as they require a pass through the entire dataset $\mathcal{D}$. For

such scenarios, SGMCMC (Welling & Teh, 2011; Chen et al., 2014; Ma et al., 2015) is a standard choice, where the gradients of the energy function $\nabla_\theta U(\theta)$ are approximated by a stochastic gradient computed from mini-batches. That is, given a mini-batch $B \subset \{1, \ldots, n\}$ where $|B| \ll n$, an unbiased estimator of the full gradient $\nabla_\theta U(\theta)$ with $B$ can be computed as

$$\nabla_\theta \tilde{U}(\theta) = -\frac{n}{|B|} \sum_{i \in B} \nabla_\theta \log p(y_i \mid x_i, \theta) - \nabla_\theta \log p(\theta). \tag{1}$$

**A complete recipe.** There may be several ways to build a Markov chain leading to the target posterior distribution. Ma et al. (2015) presented a generic recipe that includes all the convergent SGMCMC algorithms as special cases, constituting a complete framework. In this recipe, a parameter $\theta$ of interest is augmented with an auxiliary momentum variable $r$, and an Stochastic Differential Equation (SDE) of the following form is defined for a joint variable $z = (\theta, r) \in \mathbb{R}^{2d}$ as follows.

$$H(z) := U(\theta) + g(\theta, r), \quad \Gamma_i(z) := \sum_{j=1}^{2d} \frac{\partial}{\partial z_j} (D_{ij}(z) + Q_{ij}(z)),$$

$$\mathrm{d}z = [-(D(z) + Q(z)) \nabla_z H(z) + \Gamma(z)] \, \mathrm{d}t + \sqrt{2D(z)} \mathrm{d}w_t, \tag{2}$$

where $g(\theta, r)$ is the conditional energy function of the momentum $r$ such that $p(z) \propto \exp(-H(z))$ and $w_t$ is $2d$-dimensional Brownian motion. Here, $D(z) \in \mathbb{R}^{2d \times 2d}$ and $Q(z) \in \mathbb{R}^{2d \times 2d}$ are restricted to be positive semi-definite and skew-symmetric, respectively. Given this SDE, one can obtain a SGMCMC algorithm by first substituting the full gradient $\nabla_z H(z)$ with a mini-batch gradient $\nabla_z \tilde{H}(z) = \nabla_z(\tilde{U}(\theta) + g(\theta, r))$ and then discretizing it via a numerical solver such as symplectic Euler method. A notable example would be Stochastic Gradient Hamiltonian Monte Carlo (SGHMC) (Chen et al., 2014), where $g(\theta, r) = \frac{1}{2} r^\top M^{-1} r$, $D(z) = \begin{bmatrix} 0 & 0 \\ 0 & C \end{bmatrix}$, and $Q(z) = \begin{bmatrix} 0 & -I \\ I & 0 \end{bmatrix}$ for some positive semi-definite matrices $M$ and $C$, leading to an algorithm when discretized with symplectic Euler method as follows.

$$\begin{aligned} r_{t+1} &= r_t - \epsilon_t \nabla \tilde{U}(\theta_t) - \epsilon_t C M^{-1} r_t + \xi_t, \quad \xi_t \sim \mathcal{N}(0, 2C\epsilon_t) \\ \theta_{t+1} &= \theta_t + \epsilon_t M^{-1} r_{t+1}, \end{aligned} \tag{3}$$

where $\epsilon_t$ is a step-size.

The complete recipe includes interesting special cases that introduce adaptive preconditioners to improve the mixing of SGMCMC (Girolami & Calderhead, 2011; Li et al., 2016; Wenzel et al., 2020). For instance, Li et al. (2016) proposed Preconditioned Stochastic Gradient Langevin Dynamics (pSGLD), which includes RMSprop (Tieleman & Hinton, 2012)-like preconditioning matrix in the updates:

$$\begin{aligned} \theta_{t+1} &= \theta_t - \epsilon_t [G(\theta_t) \nabla_\theta \tilde{U}(\theta) + \Gamma(\theta_t)] + \xi_t, \quad \xi_t \sim \mathcal{N}(0, 2G(\theta_t)\epsilon_t) \\ V(\theta_{t+1}) &= \alpha V(\theta_t) + (1 - \alpha) \frac{\nabla_\theta \tilde{U}(\theta_t)}{n} \odot \frac{\nabla_\theta \tilde{U}(\theta_t)}{n} \\ G(\theta_{t+1}) &= \mathrm{diag}(\mathbf{1} \oslash (\lambda \mathbf{1} + \sqrt{V(\theta_{t+1})})), \end{aligned} \tag{4}$$

where $\oslash, \odot$ denotes elementwise division and multiplication, respectively. pSGLD exploits recent gradient information to adaptively adjust the scale of energy gradients and noise. However, this heuristical adjustment is still insufficient to efficiently explore the complex posteriors of BNNs (Zhang et al., 2020). Also, introducing preconditioner dependent to $\theta$ harms the computational efficiency of sampler since it requires to include additional correction term in the discretization step for correct simulation (Wenzel et al., 2020).

Recently, Zhang et al. (2020) introduced cyclic learning rate schedule for efficient exploration of multi-modal distribution. The key idea is using the spike of learning rate induced by cyclic learning rate to escape from a single mode and move to other modes. However, in our experiment, we find that SGMCMC with cyclical learning rate does not necessarily capture multi-modality and it also requires a large amount of update steps to move to other modes in practice.

**Prediction via Bayesian model averaging.** After inferring the posterior $p(\theta \mid \mathcal{D})$, for a test input $x_*$, the posterior predictive is computed as

$$p(y_* \mid x_*, \mathcal{D}) = \int_{\mathbb{R}^d} p(y_* \mid x_*, \theta) p(\theta \mid \mathcal{D}) \mathrm{d}\theta, \tag{5}$$

which is also referred to as Bayesian Model Averaging (BMA). In our setting, having collected from the posterior samples $\theta_1, \ldots, \theta_K$ from a convergent chain simulated from SGMCMC procedure, the predictive distribution is approximated with a Monte-Carlo estimater,

$$p(y_* \mid x_*, \mathcal{D}) \approx \frac{1}{K} \sum_{k=1}^{K} p(y_* \mid x_*, \theta_k). \tag{6}$$

As one can easily guess, the quality of this approximation depends heavily on the quality of the samples drawn from Markov Chain Monte Carlo (MCMC) procedure. For over-parameterzied deep neural networks that we are interested in, the target posterior $p(\theta \mid \mathcal{D})$ is typically highly multi-modal, so simple SGMCMC methods suffer from poor mixing; that is, the posterior samples collected from those methods are not widely spread throughout the parameter space, so it takes exponentially many samples to achieve desired level of accuracy for the approximation. Hence, a good SGMCMC algorithm should be equipped with the ability to efficiently explore the parameter space, while still be able to stay sufficiently long in high-density regions. That is, it should have a right balance between exploration-exploitation.

**Meta Learning** Meta-learning, or learning to learn, refers to the algorithm that learns the useful general knowledge from source tasks that can transfer to the unseen tasks. Most meta-learning algorithms involves two levels of learning: an inner-loop and outer-loop (Metz et al., 2018). Inner-loop usually contains the training procedure of particular task. In our work, inner-loop for our meta-training is iteratively update the model parameter $\theta$ by running SGMCMC with learnable transition kernel. Outer-loop refers to the training procedure of meta-parameter $\phi$, which is done by minimizing meta-objective $L(\phi)$.

As a subfield of meta-learning, learning an optimizer is emerging field which aims to learn the learnable optimizer well applied to some set of target tasks. In general, training learned optimizer includes the backpropagation through the computational graph of long inner-loop iteration. Truncated back-propagtion through time (Werbos, 1990) can be one solution, but this introduces truncation bias to the gradient estimator. Recent studies (Metz et al., 2019; 2022b) revealed that replacing back-propagation with non-analytic gradient estimation method like Evolution Strategy (ES) (Salimans et al., 2017) can improve meta-optimization. In this paper, for estimating $\nabla_\phi L(\phi)$, we do not retain computational graph to backpropagate through inner-loop.

## 3 MAIN CONTRIBUTION: LEARNING TO EXPLORE

### 3.1 META-LEARNING FRAMEWORK FOR SGMCMC

Instead of using a hand-designed recipe for SGMCMC, we aim to *learn* the proper SGMCMC update steps through meta learning. The existing works, both the methods using hand-designed choices or meta-learning (Gong et al., 2018), try to determine the forms of the matrices $D(z)$ and $Q(z)$ while keeping the kinetic energy $g(\theta, r)$ as simple Gaussian energy function, that is, $g(\theta, r) = r^\top M^{-1} r / 2$. This choice indeed is theoretically grounded, which can be shown to be optimal when the target distribution is Gaussian (Betancourt, 2017), but may not be optimal for the complex multi-modal posteriors of BNNs. We instead choose to learn $g(\theta, r)$ while keeping $D(z)$ and $Q(z)$ as simple as possible. We argue that the meta-learning approach based on this alternative parameterization is more effective in learning versatile SGMCMC procedure that scales to large BNNs.

More specifically, we parameterize the gradients of the kinetic energy function $\nabla_\theta g(\theta, r)$ and $\nabla_r g(\theta, r)$ with neural networks $\alpha_\phi(\theta, r)$ and $\beta_\phi(\theta, r)$ respectively, and set $D(z)$ and $Q(z)$ as in SGHMC. The update step of SGMCMC, when discretized with symplectic Euler method is,

$$\begin{aligned} r_{t+1} &= r_t - \epsilon_t [\nabla_\theta \tilde{U}(\theta_t) + \alpha_\phi(\theta_t, r_t) + C\beta_\phi(\theta_t, r_t)] + \xi_t, \quad \xi_t \sim \mathcal{N}(0, 2C\epsilon_t) \\ \theta_{t+1} &= \theta_t + \epsilon_t \beta_\phi(\theta_t, r_{t+1}). \end{aligned} \tag{7}$$

The neural networks $\alpha_\phi$ and $\beta_\phi$ are parameterized as two-layer Multi-Layer Perceptrons (MLPs) with 32 hidden units. Specifically, $\alpha_\phi$ and $\beta_\phi$ are applied to each dimension of parameter and momentum independently, similar to the commonly used learned optimizers (Andrychowicz et al., 2016; Metz et al., 2019). Again, following the common literature in learned optimizers (Metz et al., 2019), for each dimension of the parameter and momentum, we feed the corresponding parameter and momentum values, the stochastic gradients of energy functions for that element, and running average of the gradient at various time scales, as they are reported to encode the sufficient information about the loss surface geometry. See Appendix E for implementation details of $\alpha_\phi$ and $\beta_\phi$. By leveraging these information, we expect our meta-learned SGMCMC procedure to capture the multi-modal structures of the target posteriors of BNNs, and thus yielding a better mixing method.

### 3.2 META-OBJECTIVE AND OPTIMIZATION

**Objective functions for meta-learning.** Meta-objective should reflect the meta-knowledge one wants to learn. We design the meta-objective based on the hope that samples collected through SGMCMC should be good at approximating the posterior predictive $p(y_*|x_*, \mathcal{D})$. In order to achieve this goal, we propose the meta-objective called BMA meta-loss. After the sufficient number of inner-updates, we collect $K$ parameter samples with some interval between them (thinning). Let $\theta_k(\phi)$ be the $k^{th}$ collected parameter, and we compute the Monte-Carlo estimator of the predictive distribution and use it as a meta-objective function (note the dependency of $\theta_k$ on the meta-parameter $\phi$, as it is a consequence of learning SGMCMC with the meta-parameter $\phi$).

$$L(\phi) = -\log \frac{1}{K} \sum_{k=1}^{K} p(y_* \mid x_*, \theta_k(\phi)), \tag{8}$$

where $(x_*, y_*)$ is a validation data point.

---

**Algorithm 1:** Meta training procedure

**Input :** task distribution $P(\mathcal{T})$, inner iterations $N_{\text{inner}}$, outer interations $N_{\text{outer}}$, step size $\epsilon$, noise scale $\sigma^2$, initial meta-parameter $\phi_0$.
**Ouput:** Meta parameter $\phi$.
**for** $j = 1, \ldots, N_{\text{outer}}$ **do**
    Sample task $T_i \sim P(\mathcal{T})$
    Initialize model parameter $\theta_0$ for $T_i$
    sample $\eta \sim \mathcal{N}(0, \sigma^2 I)$
    $L(\phi + \eta) \leftarrow \text{InnerLoop}(\theta_0, \phi + \eta, \epsilon, N_{\text{inner}})$
    $L(\phi - \eta) \leftarrow \text{InnerLoop}(\theta_0, \phi - \eta, \epsilon, N_{\text{inner}})$
    $\nabla_\phi L \leftarrow \frac{1}{2\sigma^2} \eta \left( L(\phi + \eta) - L(\phi - \eta) \right)$
    $\phi \leftarrow \phi - \gamma \nabla_\phi L(\phi)$
**end**

---

**Algorithm 2:** InnerLoop

**Input :** Meta parameter $\phi$, inner interations $N_{\text{inner}}$, initial parameter $\theta_0$, step size $\varepsilon$, burn-in steps $B$, thinning interval $T$.
**Ouput:** Loss $L(\phi)$
Initialize $\Theta = \varnothing$ and $r_0 \sim \mathcal{N}(0, I_d)$.
**for** $i = 1, \ldots, N_{\text{inner}}$ **do**
    $r_{t+1} = r_t - \epsilon_t(\nabla \tilde{U}(\theta_t) + \alpha_\phi + c\beta_\phi) + \xi_t$
    where $\xi_t \sim \mathcal{N}(0, 2c)$.
    $\theta_{t+1} = \theta_t + \epsilon_t \beta_\phi$
    **if** $i > B$ & $\mod(i, T) = 0$ **then**
        $\Theta \leftarrow \Theta \cup \{\theta_i\}$
    **end**
**end**
$L(\phi) \leftarrow -\log \frac{1}{|\Theta|} \sum_{\theta \in \Theta} p(y^* | x^*, \theta)$

---

**Gradient estimation for meta-objective.** Estimating the meta-gradient $\nabla_\phi L(\phi)$ is highly non-trivial (Metz et al., 2018; 2019), especially when the number of inner update steps is large. For instance, a naïve method such as backpropagation through time would require memory grows linearly with the number of inner-steps, so become easily infeasible for even moderate sized models. One might consider using the truncation approximation, but that would result in a biased gradient estimator. Instead, we adapt ES (Salimans et al., 2017) with antithetic sampling scheme, which has been widely used in recent literature of training learned optimizer. Metz et al. (2019) showed that unrolled optimization with many inner-steps can lead to chaotic meta-loss surface and ES is capable of relieving this pathology by employing smoothed loss,

$$L(\phi) = \mathbb{E}_{\tilde{\phi} \sim \mathcal{N}(\phi, \sigma^2 I)} \left[ L(\tilde{\phi}) \right] \tag{9}$$

where $\sigma^2$ determines the degree of smoothing. Also, antithetic sampling is usually applied to reduce the estimation variance of $\nabla_\phi L(\phi)$. Through log-derivative trick, we can get unbiased estimator of

(9),

$$\hat{g} = \frac{1}{N} \sum_{i=1}^{N} L(\phi + \eta_i) \frac{\eta_i}{2\sigma^2} \quad \text{where } \eta_i \overset{\text{i.i.d}}{\sim} \mathcal{N}(0, \sigma^2 I), \; i \in \{1 \ldots N\} \tag{10}$$

. In addition, we can get another unbiased estimator $\hat{g}^{-1} = -\frac{1}{N} \sum_{i=1}^{N} L(\phi - \eta_i) \frac{\eta_i}{2\sigma^2}$ by reusing the negative of $\eta_i$. By taking the average of two estimators, we can obtain the following gradient estimator.

$$\nabla_\phi \hat{L}(\phi) = \frac{1}{N} \sum_{i=1}^{N} \left[ \frac{L(\phi + \eta_i) - L(\phi - \eta_i)}{2\sigma^2} \right] \eta_i \tag{11}$$

The estimator is also amenable to parallelization, improving the efficiency of gradient computation.

### 3.3 META TRAINING PROCEDURE

**Generic pipeline.** General process of meta-training is as follows. First, for each inner-loop, we sample a task from the pre-determined task distribution. An inner-loop starts from an randomly initialized parameter and iteratively apply update step (7) to run a single chain of SGMCMC. In the initial stage of meta-training, the chains from these inner loops show poor convergence, but the performance improves as training progresses. Similar to general Bayesian inference, we consider the early part of the inner loop as a burn-in period and collect samples from the end of the inner-loop at regular intervals when evaluating the meta-objective. This training process naturally integrates the meta-learning and Bayesian inference in that mimicking the actual inference procedure of Bayesian methods in realistic supervised learning tasks. In Figure 2 we show that L2E achieve desired level of accuracy for the approximation of posterior predictive with relatively small number of samples. This result indicates that L2E has successfully acquired the desired properties through meta-training.

**Multitask training for better generalization.** In meta-learning, it is commonly known that diversifying the task distribution helps to improve generalization performance. We include various neural network architectures and datasets in the task distribution to ensure that L2E has sufficient generalization capacity. Also, we evaluate how the diversity of the task distribution affects the performance of L2E in Table 12.

## 4 EXPERIMENTS

In this section, we will evaluate the performance of L2E in various aspects. Through extensive experiments, we would like to demonstrate followings:

- L2E shows remarkable performance both on real-world image classification task and Out-of-Distribution (OOD) detection task comparing to other competitive baseline methods.

- L2E can effectively sample from BNNs posterior distribution, collecting diverse set of parameters both in weight space and function space.

**Experimental details** For image classification experiments, we compare L2E with Deep Ensembles (DE), Cyclical Stochastic Gradient MCMC (CSGMCMC) and Preconditioned Cyclic Stochastic Gradient MCMC (P-CSGMCMC). We use 1 layer convolution neural network on fashion-MNIST (Xiao et al., 2017), ResNet20 on CIFAR-10, CIFAR-100 (Krizhevsky et al., 2009) and ResNet56 on Tiny-ImageNet (Le & Yang, 2015). We set total training epochs for all methods to a similar level. Please refer to Appendix G.1 for experimental setup and hyperparameter settings. We report Accuracy (ACC), Negative Log-Likelihood (NLL), Expected Calibration Error (ECE) (Naeini et al., 2015) and pairwise Kullback–Leibler divergence (KLD) between probabilistic outputs from different parameters on test dataset. Throughout experiments, we do not use data augmentation since it produces significant modifications to the likelihood function so that we cannot interpret this as a valid likelihood function(Wenzel et al., 2020). We report the mean and standard deviation of results over three different trials.

Table 1: Image classification results.

| Dataset | Method | ACC ↑ | NLL ↓ | ECE ↓ | KLD ↑ |
|---|---|---|---|---|---|
| Fashion-MNIST | DE | **0.915**±0.001 | **0.243**±0.000 | 0.009±0.001 | 0.110±0.004 |
| | CSGMCMC | 0.911±0.002 | 0.277±0.008 | 0.021±0.002 | 0.080±0.004 |
| | P-CSGMCMC | 0.912±0.001 | 0.254±0.003 | **0.005**±0.000 | 0.062±0.004 |
| | L2E | **0.917**±0.002 | **0.245**±0.002 | 0.008±0.001 | **0.247**±0.000 |
| CIFAR-10 | DE | 0.893±0.002 | 0.327±0.002 | 0.032±0.001 | 0.497±0.002 |
| | CSGMCMC | 0.884±0.002 | 0.374±0.004 | 0.065±0.004 | 0.490±0.004 |
| | P-CSGMCMC | 0.875±0.001 | 0.393±0.005 | **0.049**±0.001 | **0.726**±0.012 |
| | L2E | **0.904**±0.001 | **0.307**±0.004 | 0.053±0.001 | 0.648±0.002 |
| CIFAR-100 | DE | 0.679±0.003 | 1.231±0.004 | 0.144±0.003 | **1.638**±0.006 |
| | CSGMCMC | 0.675±0.002 | 1.234±0.006 | 0.132±0.005 | 1.413±0.003 |
| | P-CSGMCMC | 0.580±0.005 | 1.526±0.016 | **0.079**±0.006 | 0.948±0.045 |
| | L2E | **0.697**±0.001 | **1.135**±0.002 | 0.130±0.001 | 1.466±0.002 |
| Tiny-ImageNet | DE | **0.583**±0.003 | 1.783±0.015 | 0.109±0.003 | **1.936**±0.005 |
| | CSGMCMC | 0.562±0.002 | 1.811±0.004 | 0.096±0.002 | 0.776±0.009 |
| | P-CSGMCMC | 0.457±0.009 | 2.273±0.007 | **0.038**±0.002 | 1.178±0.003 |
| | L2E | **0.585**±0.003 | **1.740**±0.001 | 0.111±0.001 | 1.075±0.003 |

Table 2: CIFAR-10-C results.

| Level / Method | 1 | 2 | 3 | 4 | 5 |
|---|---|---|---|---|---|
| DE | 0.869 | 0.853 | **0.839** | **0.822** | **0.794** |
| CSGMCMC | 0.855 | 0.837 | 0.823 | 0.803 | 0.774 |
| L2E | **0.874** | **0.854** | 0.839 | 0.817 | 0.784 |

Table 3: OOD detection AUROC.

| In-dist | OOD | DE | CSGMCMC | L2E |
|---|---|---|---|---|
| CIFAR-10 | CIFAR-100 | 0.836±0.001 | 0.828±0.003 | **0.851**±0.001 |
| | SVHN | 0.919±0.001 | 0.906±0.010 | **0.931**±0.003 |
| | Tiny-ImageNet | 0.844±0.001 | 0.833±0.003 | **0.853**±0.001 |
| CIFAR-100 | CIFAR-10 | **0.793**±0.001 | 0.791±0.005 | 0.792±0.001 |
| | SVHN | 0.917±0.002 | 0.904±0.002 | **0.917**±0.003 |
| | Tiny-ImageNet | 0.780±0.001 | 0.775±0.001 | **0.782**±0.003 |

**Meta-training details** We construct set of meta-training tasks using various datasets and model architectures. Specifically, we use MNIST, Fashion-MNIST, EMNIST (Cohen et al., 2017) and MedMNIST (Yang et al., 2021) as meta-training datasets. For model architecture, we fix the general structure with several convolution layers followed by readout MLP layer. For each outer training iteration, we randomly choose dataset and sample the configuration of architecture including number of channels, depth of the convolution layers and whether to use skip connections. See Appendix E for detailed configuration of task distribution. For evaluation, we use same meta-parameter of L2E for all experiments to check the generalization ability of L2E.

## 4.1 REAL-WORLD IMAGE CLASSIFICATION

Table 1 shows the results of image classifcation experiments. We confirm that L2E outperforms other baselines in terms of predictive accuracy in general. Specifically, only DE shows comparable predictive accuracy comparing to L2E in some experiments. Among datasets for evaluation, *fashion-MNIST is the only dataset included in our task distribution.* Despite not having seen other datasets during meta-training, L2E consistently outperforms other tuned baseline methods. This clearly shows that L2E can scale and generalize well to unseen problems. In terms of functional diversity of ensemble, L2E consistently shows competitive performance among baselines. Note that KLD should be considered along with predictive accuracy since functional diversity usually declines when the predictive error of individual members is reduced (Fort et al., 2019). Taking this into account, L2E clearly outperforms similar or better level of diversity comparing to other SGMCMC methods even though predictive accuracy of L2E is the best among baselines. In complex datasets like CIFAR-100 and Tiny-ImageNet, we can find that functional diversity of DE, which necessarily captures multiple modes in the loss surface, is significantly better than other methods. Also, we confirm that introducing preconditioner to SGMCMC does not necessarily improve general performance of SGMCMC. P-CSGMCMC shows comparable performance with CSGMCMC in Fashion-MNIST dataset, but it significantly underperforms in other experiments.

## 4.2 OUT-OF-DISTRIBUTION(OOD) DETECTION

Bayesian methods are frequently used for OOD detection task. In Table 3, we report the OOD detection performance of baseline methods and L2E. We use Maximum Softmax Probability (MSP) which is equivalent to confidence of logit as OOD score. Difference of confidence between in-distribution data and OOD data is measured using Area Under the ROC curve (AUROC) (Liang et al., 2017). For Tiny-ImageNet, we resize the image to 32×32. Firstly, among models trained on CIFAR-10, L2E shows the best performance for all OOD datasets. For models trained on CIFAR-100, L2E is significantly better than CSGMCMC and comparable to DE. Since DE is a very strong baseline in uncertainty estimation, we can confirm that L2E is competitive in OOD detection.

Next, we consider CIFAR-10-C (Hendrycks & Dietterich, 2019) for evaluating robustness to covariate shift. We use accuracy on corrupted data for the metric. Table 2 shows that L2E is more robust to covariate shift than CSGMCMC for all levels of corruption. However, with intensive corruption, L2E

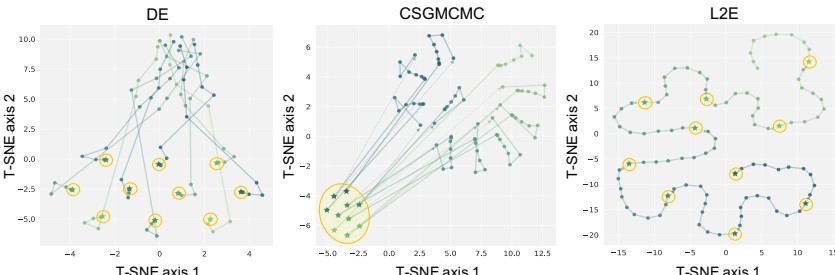

Figure 3: $t$-SNE visualization of predictions from different model parameters of DE, CSGMCMC and L2E on CIFAR-10 test dataset. Highlighted points represent parameters which are collected for BMA. L2E covers almost the entire space despite using a single trajectory.



Figure 4: Loss surface of ResNet56 on Tiny-ImageNet as a function of model parameters in a 2-dimensional subspace spanned by solutions of DE, CSGMCMC, P-CSGMCMC and L2E. Colors represent the level of test accuracy. *Left* and *Right* plots clearly display the multi-modality while *Middle* plot does not.

exhibits lower accuracy than DE. This aligns with the results from Izmailov et al. (2021a) that BNN are not robust to covariate shift in reality. Nevertheless, L2E still demonstrates better performance than CSGMCMC, implying the advantage over CSGMCMC when using BNN under covariate shift.

### 4.3 L2E CAN CAPTURE MULTI-MODALITY

In Figure 3 and Figure 4, we can observe the behavior of DE, CSGMCMC, and L2E in function space. Firstly, we save the model parameters with short intervals (5 epochs) and visualize them along with their predictions using t-SNE (Van der Maaten & Hinton, 2008) in Figure 3. Notably, L2E exhibits a distinctive pattern that appears to traverse a variety of areas within the space. In Figure 4, we display the loss surface using a 2-dimensional subspace spanned by the first three collected parameters for each method following Garipov et al. (2018). Parameters of DE clearly located on multiple distinct modes as expected. In contrast, CSGMCMC seems to sample parameters within a single mode, while samples from L2E appear to be in distinct modes. For deeper investigation, we plot test error along a linear path between multiple pairs of saved parameters inspired by Goodfellow et al. (2014). If there is a loss barrier between the two parameters, indicating that they belong to different modes, classification error significantly increases along the linear path between two parameters. In Figure 6, L2E shows a significant increase in predictive error along the linear path between every pair of parameters while CSGMCMC exhibits a relatively low level of the loss barrier between samples. This suggests that L2E is capable of the capturing multi-modality of the posterior distribution.

CSGMCMC attempts to explore multi-modality by exploiting artificial spikes induced by the learning rate schedule. Since it inevitably deviates from high-density regions for exploration, it requires a sufficient number of update steps to return to the high-density region. In contrast, L2E shows better exploration-exploitation balance than CSGMCMC because it continuously explores various modes while staying in high-density regions without using a learning rate schedule. In addition, in Figure 19, CSGMCMC shows a significant decrease in predictive accuracy and diversity as the thinning interval gets shorter while L2E can maintain a similar level of accuracy and diversity even with a shorter thinning interval, making it a much more computationally efficient approach in practice.

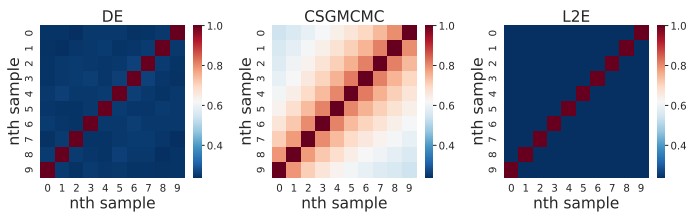 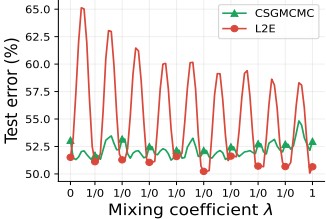

Figure 5: Cosine similarity between weights of ResNet56 on Tiny-ImageNet. DE and L2E collect more diverse solutions in weight space than CSGMCMC.

Figure 6: Test error along linear path between a pair of parameters

Table 4: ESS / wall clock time.

| Method / Dataset | CSGMCMC | P-CSGMCMC | L2E |
|---|---|---|---|
| Fashion-MNIST | $219.85_{\pm 6.64}$ | $526.73_{\pm 5.42}$ | $136.31_{\pm 0.42}$ |
| CIFAR-10 | $45.73_{\pm 0.22}$ | $108.90_{\pm 0.18}$ | $75.91_{\pm 0.15}$ |
| CIFAR-100 | $33.62_{\pm 0.12}$ | $114.12_{\pm 0.01}$ | $73.54_{\pm 0.48}$ |
| Tiny-ImageNet | $1.93_{\pm 0.12}$ | $1.70_{\pm 0.01}$ | $1.71_{\pm 0.00}$ |

Table 5: Proportion of samples with $\hat{R} < 1.2$.

| Method | Fashion-MNIST | CIFAR-10 | CIFAR-100 | Tiny-ImageNet |
|---|---|---|---|---|
| CSGMCMC | 0.238 | 0.542 | 0.524 | 0.600 |
| P-CSGMCMC | 0.898 | 0.872 | 0.722 | 0.803 |
| L2E | **0.992** | **0.953** | **0.800** | **0.880** |

### 4.4 CONVERGENCE ANALYSIS

To evaluate whether L2E converge to the target distribution, we use $\hat{R}$ (Gelman & Rubin, 1992). $\hat{R}$ compares the variance between multiple chains to the variance within a single chain. If it is significantly greater than 1.0, it implies the poor mixing of chains. While the desirable level of $\hat{R}$ is problem-specific, we use the criterion proposed by Brooks & Gelman (1998), $\hat{R} < 1.2$, to evaluate the degree of mixing. In Table 5, we report the proportion of parameters with $\hat{R}$ values less than 1.2. Overall, L2E method demonstrates good mixing, with over 95% of parameters showing $\hat{R} < 1.2$ in Fashion-MNIST and CIFAR-10 experiments. L2E consistently demonstrates descent performance in various experiments, whereas CSGMCMC exhibits poor mixing across all experiments, indicating that CSGMCMC fails to sample from BNNs posterior distribution. In general, applying preconditioner seems to improve mixing, but it is evident that the predictive performance of P-CSGMCMC is still significantly suboptimal.

Also, we measure Effective Sample Size (ESS) to quantify the quality of the samples in terms of their independence and effectiveness in representing the underlying distribution. We report ESS normalized with respect to wall clock time of each method to evaluate the sampling efficiency in fixed computational cost. In Table 4, P-CSGMCMC achieves better normalized ESS than other methods except for Tiny-ImageNet experiment. However, its poor predictive performance diminishes the preference for P-CSGMCMC. Taking this into account, L2E demonstrates comparable level of efficiency among practical methods, even utilizing additional neural networks. We also report wall clock time of each method in Table 18.

## 5 CONCLUSION

In this work, we introduced a novel meta-learning framework called L2E to improve SGMCMC methods. Unlike conventional SGMCMC methods that heavily rely on manually designed components inspired by mathematical or physics principles, we aim to learn critical design components of SGMCMC directly from data. Through experiments, we show numerous advantages of L2E over existing SGMCMC methods, including better mixing, improved prediction performance, and a decreased need for tuning hyperparameters. Our learning-based approach would be a promising direction to solve several challenges that SGMCMC methods face in BNNs.

**Ethics and Reproducibility statement** Please refer to Appendix G for full experimental detail including datasets, models, and evaluation metrics. We have read and adhered to the ethical guideline of International Conference on Learning Representations in the course of conducting this research.

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

Table 6: Comparison between Gong et al. (2018) and L2E

| | Gong et al. (2018) | L2E |
|---|---|---|
| Purpose of meta-learning | Fast convergence, low bias | Efficient exploration of multi-modal BNNs posterior |
| Learning target | Diffusion and curl matrix | Gradient of kinetic energy |
| Meta-training task | Single-task | Multi-task |
| Meta-objective | $\mathrm{KL}(q_t\|\pi)$ | $-\log \frac{1}{K}\sum_{k=1}^{K} p(y_*\,|\,x_*, \theta_k(\phi))$ |
| Meta-gradient estimation | TBPTT (Werbos, 1990) | ES (Salimans et al., 2017) |
| Generalizes to | unseen classes in same dataset | completely different dataset |
| Scales to | small scale architecture | large scale architecture |

Paul J Werbos. Backpropagation through time: what it does and how to do it. *Proceedings of the IEEE*, 78(10):1550–1560, 1990. 4, 12

Han Xiao, Kashif Rasul, and Roland Vollgraf. Fashion-mnist: a novel image dataset for benchmarking machine learning algorithms. *arXiv preprint arXiv:1708.07747*, 2017. 6

Jiancheng Yang, Rui Shi, and Bingbing Ni. Medmnist classification decathlon: A lightweight automl benchmark for medical image analysis. In *2021 IEEE 18th International Symposium on Biomedical Imaging (ISBI)*, pp. 191–195. IEEE, 2021. 7

Ruqi Zhang, Chunyuan Li, Jianyi Zhang, Changyou Chen, and Andrew Gordon Wilson. Cyclical stochastic gradient mcmc for bayesian deep learning. In *International Conference on Learning Representations (ICLR)*, 2020. 1, 3, 16, 20

# A   COMPARISON WITH GONG ET AL. (2018)

In Table 6, we demonstrate the difference between Meta-SGMCMC (Gong et al., 2018) and L2E. Meta-SGMCMC aims to build a sampler through meta-learning that rapidly converges to the target distribution and performs accurate simulation. This goal aligns with the objectives of all SGMCMC methods. However, our approach is specifically designed with the more concrete purpose of effectively simulating multi-modal DNN posterior distribution and also generalizing to unseen problems. We compare L2E to Meta-SGMCMC in various aspects in the following subsections.

## A.1   DIFFERENCE IN PARAMETERIZATION OF META MODELS

To achieve our goal, we did not follow the design of the diffusion matrix and curl matrix as done by Meta-SGMCMC. We firstly state the update rule for a joint variable $z = (\theta, r) \in \mathbb{R}^{2d}$ presented in Gong et al. (2018).

$$
\begin{aligned}
r_{t+1} &= (1 - \epsilon_t D_f(z_t))r_t - \epsilon_t Q_f(z_t)\nabla_\theta \tilde{U}(\theta_t) + \epsilon_t \Gamma_r(z_t) + \xi_t \\
\theta_{t+1} &= \theta_t + \epsilon_t Q_f(z_t)r_t + \epsilon_t \Gamma_\theta(z_t) \\
\xi_t &\sim \mathcal{N}(0, 2\epsilon_t D_f(z_t)), \quad \Gamma_i(z) := \sum_{j=1}^{2d} \frac{\partial}{\partial z_j}(D_{ij}(z) + Q_{ij}(z))
\end{aligned}
\tag{12}
$$

They aimed to learn $D_f$ and $Q_f$ to build SGMCMC that can fastly converge to target distribution with small bias. This design has two limitations to achieve our goal, efficient exploration of multi-modal large scale BNNs posterior distribution. Firstly, learning $D_f$ and $Q_f$ to be dependent on $z$ harms scalability of algorithm since it introduces a new correction term $\Gamma_i(z)$ which forces to compute the gradient of $D_f$ and $Q_f$ with respect to $z$. This makes significant computational burden as the dimension of $z$ increases, and requires an additional approximation. While Meta-SGMCMC attempted to address this issue of computational cost through finite difference, but it still introduces additional computation with time complexity $O(HD)$ where $H$ is number of hidden units of neural network in meta-sampler and $D$ is dimension of $\theta$. Also, it is important to note that adding such

Table 7: Experiments on CIFAR-10 following experiments in Gong et al. (2018)

| Methods | NT ACC | NT+AF ACC | NT+Data ACC | NT NLL/100 | NT+AF NLL/100 | NT+Data NLL/100 |
|---|---|---|---|---|---|---|
| Meta-SGMCMC | $78.12_{\pm0.035}$ | $74.41_{\pm0.11}$ | $89.97_{\pm0.04}$ | $68.88_{\pm0.15}$ | $79.55_{\pm0.057}$ | $15.66_{\pm0.28}$ |
| L2E | $\mathbf{79.21}_{\pm0.203}$ | $\mathbf{75.91}_{\pm0.200}$ | $\mathbf{92.49}_{\pm0.234}$ | $\mathbf{63.23}_{\pm0.46}$ | $\mathbf{72.11}_{\pm0.22}$ | $\mathbf{11.82}_{\pm0.81}$ |

Table 8: Experiments on MNIST following experiments in Gong et al. (2018)

| Methods | NT ACC | NT+AF ACC | NT+Data ACC | NT NLL/100 | NT+AF NLL/100 | NT+Data NLL/100 |
|---|---|---|---|---|---|---|
| Meta-SGMCMC | $98.36_{\pm0.02}$ | $97.72_{\pm0.02}$ | $98.62_{\pm0.02}$ | $640_{\pm6.25}$ | $875_{\pm3.19}$ | $230_{\pm3.23}$ |
| L2E | $\mathbf{98.39}_{\pm0.05}$ | $\mathbf{98.07}_{\pm0.08}$ | - | $\mathbf{558}_{\pm3.19}$ | $\mathbf{679}_{\pm6.65}$ | - |

approximation can negatively impact the convergence of the sampler. On the other hand, as we keep the diffusion and curl matrices independent of $z$ and learns the kinetic energy, we can avoid this pathology without sacrificing the flexibility of the sampler.

Additionally, since $D_f$ and $Q_f$ mainly function as multipliers for gradient and momentum, learning them may not be as effective as it should be for effective exploration in low energy regions. In low energy regions where the norm of gradient and momentum are extremely small, it is difficult to make reasonable amount of update of $\theta$ for exploration by multiplying $Q_f$ to momentum and gradient. By contrast, In the update rule of L2E (7), $\alpha_\phi, \beta_\phi$ are added to the gradient, which is more suitable for controlling the magnitude and direction of update in low energy regions. Also, ablation study in Table 12 shows the inferior performance of learning $D_f$ and $Q_f$ in terms of classification accuracy and predictive diversity in our setting. Therefore, we argue that learning kinetic energy gradient is better than learning $D_f$ and $Q_f$, as it avoids additional computation and allows the sampler to mix better especially around the low energy region.

## A.2 DIFFERENCE IN META-TRAINING PROCEDURE

There are significant differences between two methods in meta-training pipeline. Firstly, the most notable distinction is that Meta-SGMCMC learns with only one task, requiring the training of a new learner for each problem. This poses an issue as a single learner may not generally apply well to various problems. In contrast, our approach involves sampling from a task set composed of diverse datasets and architectures for training. Another crucial difference is the choice of estimator for the meta-objective gradients. Gong et al. (2018) employs Truncated BackPropagation Through Time (TBPTT) which truncates computational graphs of a long inner-loop to estimate the meta gradient. While this approach can save computational cost by avoiding backpropagation through long computational graph, it results in a biased estimator of meta-gradient. To address these issues, we utilize ES to compute unbiased estimator of meta-gradient efficiently.

## A.3 DIFFERENCE IN META-TRAINING OBJECTIVES

Previous studies have proposed various meta-objectives to achieve goals similar to ours. Gong et al. (2018) minimizes the $KL(q_t|\pi)$, where $\pi$ is target distribution and $q_t$ is the marginal distribution of $\theta$ at time $t$ for good mixing. On the other hand, Levy et al. (2017) employs meta-objective maximizing the jump distances between samples in weight space and simultaneously minimizing the energy in order to make sampler rapidly explore between modes. However, explicitly maximizing the jump distances in weight space can be easily cheated, as the distances between weights does not necessarily lead to the difference in the functions, resulting in trivial sampler with which achieving the balance between convergence and exploration is hard. Also, minimizing the divergence with target distribution seems sensible, but due to intractable $q_t$, computing the gradient of $q_t$ should resort to gradient estimator. Gong et al. (2018) used stein-gradient estimator, which requires multiple independent chains so it harms scalability. Also, this objective does not lead the learned sampler to explore multi-modal distribution. Gong et al. (2018, Figure 3) shows that the learned sampler

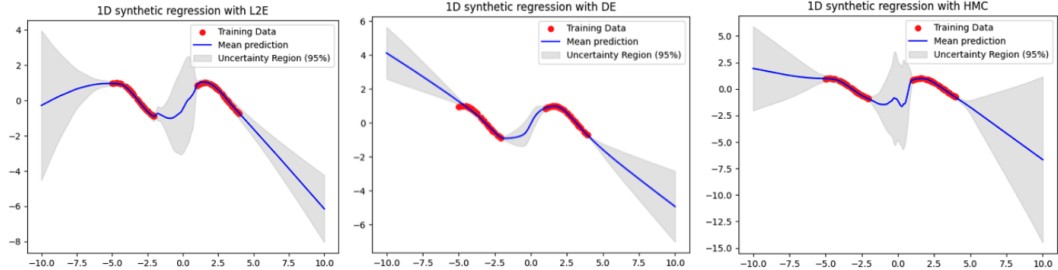

Figure 7: 1-D synthetic regression of L2E, DE, HMC. See § 5 for details and discussion.

quickly converges to low energy region, but learned friction term $D_f$ restricts the amount of update in low energy region, limiting the exploration behaviour. Among choices, we find out that BMA meta-loss is a simple yet effective meta-objective that naturally encodes exploration-exploitation balance without numerical instability and exhaustive hyperparameter tuning.

### A.4    EXPERIMENTAL RESULTS

In order to compare L2E with Meta-SGMCMC, we exactly replicate the experimental setup of MNIST and CIFAR-10 experiments in Gong et al. (2018) and evaluate performance of L2E. These experiments evaluate how well each method generalizes to unseen Neural Network architecture (NT), activation function (AF) and dataset (Data) which were unseen during the meta-learning process. For the experimental details, please refer to the experiments section of Gong et al. (2018). We use same scale of metrics in Gong et al. (2018) for conveniently comparing two methods. We report ACC and NLL on test dataset. Since L2E used MNIST dataset during meta-train, we do not evaluate Dataset generalization experiments on MNIST. Meta-SGMCMC used 20 parallel chains with 100 epochs for MNIST and 200 epochs for CIFAR-10. We use single chain with 100 burn-in epochs for both experiments, and use 10 thinning epochs for collecting 20 samples.

In Table 7 and Table 8, we confirm that L2E significantly outperforms Meta-SGMCMC for all generalization types in terms of ACC and NLL. Notably, in CIFAR-10 experiment, despite L2E was not trained on the CIFAR-10 during meta-learning, L2E significantly outperforms Meta-SGMCMC with a wide margin indicating that our approach better generalizes to unseen datasets compared to Meta-SGMCMC

## B    ANALYSIS OF EXPLORATION PROPERTY

In this section, we analyze how L2E can collect diverse samples in terms of weight space and function space with a single MCMC chain. We plot $l2$ norm of $\Delta\theta = \theta_{t+1} - \theta_t$ at time $t$, and training cross-entropy loss (NLL) for 200 epochs and comparing it with DE and CSGMCMC. For L2E, this is equivalent to track the norm of parameterized gradient $\beta_\phi$ in (7) since $||\beta_\phi||^2 = \frac{||\Delta\theta||^2}{\epsilon^2}$. In Figure 8, we find that L2E updates $\theta$ with a larger magnitude in local minima than in the early stages of training. This tendency is different from other gradient-based optimizer or MCMC methods where the amount of update is relatively small at local minima. Additionally, we notice that L2E actively updates $\theta$ at minima while maintaining loss as nearly constant. This trend is consistently observed in both CIFAR-10 and CIFAR-100, implying that L2E learns some common knowledge of posterior information across tasks for efficient exploration in low loss regions. Various experimental results (e.g., see Figures 3 to 6) support that L2E is good at capturing multi-modalities of BNNs posterior with a single trajectory. Since L2E produces significant amount of updates at local minima without increasing the loss, we can say that our parameterized gradients learned the general knowledge to explore high density regions among different modalities.

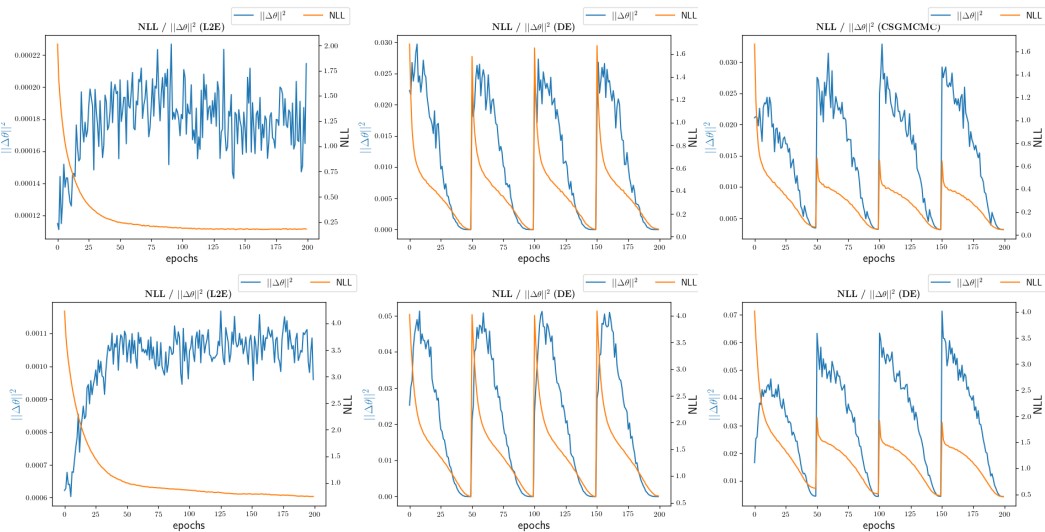

Figure 8: Plots of $||\Delta\theta||^2$ and train NLL during training of L2E, DE, CSGMCMC. Top: CIFAR-10, Below: CIFAR-100. L2E updates $\theta$ with a larger magnitude in local minima than in the early stages of training while other methods only make very small updates of $\theta$ at local minima.

## C   ADDITIONAL EXPERIMENTS

### C.1   1-D SYNTHETIC REGRESSION

We conduct 1-D synthetic regression task to visually check whether L2E can capture the epistemic uncertainty. For the training data, we generate 1000 data points from underlying true function $y = \sin(x)$, within the interval $[-5, 1]$ and $[1, 4]$. We use DE and HMC as baselines. We collect 50 parameters for each methods and plot the mean prediction and $95\%$ confidence interval of the prediction. For L2E, we do not fine-tune the learned sampler used in the main experiments. We use thinning interval of 50 training steps and 1000 burn-in steps for L2E, 300 training steps for each single solution of DE and 1000 burn-in steps and 100 leap-frog steps for HMC. We use 2 layers MLP with 100 hidden units and ReLU activation to estimate the function.

Effective method for capturing epistemic uncertainty should make confident predictions for the training data and should be uncertain on OOD data points. Figure 7 shows that L2E better captures 'in-between' uncertainty than DE. L2E generally produces diverse predictions for out-of-distribution data points, especially for the input space between two clusters of training data while DE shows relatively confident prediction in that region. HMC is known for the golden standard for posterior inference in Bayesian method. Our experimental result aligns with this common knowledge since HMC is the best in terms of capturing epistemic uncertainty among three methods especially in areas out of the range of training points. While our approach falls short of HMC, it demonstrates better uncertainty estimation than that of DE and shows similar predictive uncertainty with HMC between two training points cluster. These experimental results align with the OOD detection experiments and convergence diagnostics presented in our main text, indicating L2E performs effective posterior inference. It is important to note that even without meta-training on the regression datasets, L2E can adapt well to the regression problem.

### C.2   IMAGE CLASSIFICATION WITH DATA AUGMENTATION

Since applying data augmentation violates Independently and Identically Distributed(IID) assumption of the dataset which is commonly assumed by Bayesian methods, this can lead to model mis-

specification (Wenzel et al., 2020; Kapoor et al., 2022). Therefore, prior work such as Izmailov et al. (2021b) argued the incompatibility between Bayesian methods and data augmentation. However, data augmentation is an indispensable technique in modern machine learning so it is also interesting to see how L2E performs with data augmentation. We run the image classification experiments on CIFAR-10 and CIFAR-100 with random crop and horizontal flip for the data augmentation. Since using data augmentation introduces Cold Posterior Effect (Wenzel et al., 2020) for Bayesian methods such as CSGMCMC and L2E, we additionally tune the temperature hyperparameter for these methods. We use $T = 0.01$ for CSGMCMC and $T = 0.001$ for L2E in both experiments. Please refer to Wenzel et al. (2020) for detailed analysis of Cold Posterior Effect. We collect 10 samples for all methods for these experiments.

In Table 9, in terms of predictive accuracy, DE outperforms other baselines on CIFAR-10 and L2E beats other baselines on CIFAR-100. With data augmentation, it is not very surprising that DE outperforms other Bayesian methods like L2E and CSGMCMC in terms of predictive accuracy and calibration since they suffer from model misspecification and temperature tuning can partially handle this problem (Kapoor et al., 2022). This can explain why L2E shows worst NLL in CIFAR-10 experiment. When it comes to predictive diversity, L2E significantly outperforms baselines on both experiments. Although applying data augmentation introduces significant variations to the posterior landscape, we confirm that L2E still maintains the exploration property. To sum up, we argue that L2E is still practical method even with data augmentation since it shows competitive predictive performance and efficiently explores the posterior landscape.

### C.3 ADDITIONAL ABLATION STUDIES

In this section, we conduct additional ablation studies to check how length of inner-loops affect the convergence of the learned sampler, and whether diversity of meta-train datasets affect OOD detection performance. We use two variants of L2E for each experiment.

- **Short L2E**: meta-trained with 1000 inner-loop steps. L2E originally uses 3000 inner-loop steps for meta-training.
- **Small L2E**: meta-trained only on Fasion-MNIST dataset.

Table 11 shows that model meta-trained with short inner-loop length(Short-L2E) makes significant drop of predictive accuracy and functional diversity. This implies that length of inner-loop is crucial for learning to explore and converge. When inner loop is too short, the sampler may not sufficiently learn information about the loss surface in low-loss regions during meta-training. Then, the sampler will fail to learn desirable properties such as exploration in minima or fast convergence.

In Table 10, we confirm that seeing more datasets during meta-training stage do not necessarily lead to better OOD detection performance. Small L2E shows almost similar OOD detection performance with L2E or even better performance for some datasets.

## D  RELATED WORKS

There are three lines of literature related to our work.

**Scalable MCMC**   Since the scale of modern machine learning tasks get bigger in terms of model size and dataset size, Welling & Teh (2011) presented scalable MCMC algorithm and this becomes popular for posterior inference in BNNs. However, in real world application, these algorithms suffer from slow convergence and have difficulty with sampling from multi-modal posterior distribution since they requires small stepsizes to converge to target distribution. To improve sampling efficiency of SGMCMC, several variants have been proposed. Chen et al. (2014) introduced additional momentum variable to promote fast traversing, Ma et al. (2015) and Li et al. (2016) introduced preconditioner to exploit the local geometry of target distribution. Recently, Zhang et al. (2020) proposed CSGMCMC, integrating SGMCMC and cyclic learning rate for exploring diverse modes of posterior distribution. This algorithm shows competitive performance in terms of predictive accuracy, but still insufficient for capturing multi-mode in BNNs (Fort et al., 2019).

Table 9: Results of image classification with data augmentation.

| Dataset | Method | ACC ↑ | NLL ↓ | ECE ↓ | KLD ↑ |
|---|---|---|---|---|---|
| CIFAR-10 | DE | **0.927**±0.001 | **0.218**±0.002 | 0.021±0.001 | 0.238±0.001 |
| | CSGMCMC | 0.923±0.001 | 0.239±0.003 | **0.014**±0.001 | 0.125±0.001 |
| | L2E | 0.924±0.003 | 0.249±0.001 | 0.042±0.002 | **0.332**±0.003 |
| CIFAR-100 | DE | 0.707±0.001 | **1.053**±0.002 | 0.096±0.005 | 0.515±0.002 |
| | CSGMCMC | 0.697±0.001 | 1.066±0.007 | 0.064±0.001 | 0.373±0.005 |
| | L2E | **0.710**±0.001 | **1.053**±0.005 | **0.053**±0.001 | **0.911**±0.007 |

Table 10: Ablation study on OOD detection

| In-dist | OOD | small L2E | L2E |
|---|---|---|---|
| CIFAR-10 | CIFAR-100 | **0.855**±0.001 | 0.851±0.001 |
| | SVHN | 0.929±0.001 | **0.931**±0.003 |
| | Tiny-ImageNet | **0.855**±0.001 | 0.853±0.001 |
| CIFAR-100 | CIFAR-10 | 0.789±0.003 | **0.792**±0.002 |
| | SVHN | 0.912±0.004 | **0.917**±0.003 |
| | Tiny-ImageNet | 0.777±0.001 | **0.782**±0.003 |

Table 11: Results of ablation study for inner-loop length

| Dataset | L2E | ACC ↑ | NLL ↓ | ECE ↓ | KLD ↑ |
|---|---|---|---|---|---|
| CIFAR-10 | Short L2E | 0.8726 | 0.4021 | 0.0640 | 0.3661 |
| | L2E | 0.8952 | 0.3310 | 0.0413 | 0.6296 |
| CIFAR-100 | Short L2E | 0.6257 | 1.3811 | 0.1186 | 0.9404 |
| | L2E | 0.6737 | 1.2205 | 0.0962 | 1.5836 |

**Learned optimization**   Recently, there have been several studies replacing hand-designed optimizers such as Adam and SGD by learnable function parameterized by Neural Networks. Andrychowicz et al. (2016) firstly demonstrated that meta-learning the update functions can outperform hand-designed optimizers in modern machine learning tasks. Recently, Metz et al. (2022b) released general purpose learned optimizer, which is meta-trained on set of small scale machine learning problems and can generalize to many unseen large scale machine learning tasks.

**Meta-Learning and MCMC**   There exists line of work that parameterize the transition kernel of MCMC with trainable function for various purposes. Levy et al. (2017) and Song et al. (2017) used learnable invertible operator to automatically design the transition kernel of HMC for good mixing. For SGMCMC, Gong et al. (2018) parameterized the curl matrix and acceleration matrix using neural networks under the framework of Ma et al. (2015). Gong et al. (2018) is the closest work for our method. We demonstrate the difference with this work and other discussions in § 5.

# E   DETAILS FOR META-TRAINING

For meta-training, we construct our experiment code based on JAX learned-optimization package (Metz et al., 2022a). Also, we build our own meta-loss and L2E with specific input features and design choice. Therefore, we construct our own meta-learning task distribution that L2E can efficiently learn knowledge for better generalization for large-scale image classification task.

## E.1   INPUT FEATURES

We use the following input features for L2E:

- raw gradient values
- raw parameter values
- raw momentum values
- running average of gradient values

Running average feature is expanded for multiple time scale in that we use multiple momentum-decay values for averaging. We use 0.1, 0.5, 0.9, 0.99, 0.999 and 0.9999 for momentum decay so that running average feature is expanded into 6-dimensions. Therefore, we have total 9-dimensional input features for each dimension of parameter and momentum. Input features are normalized so that $l2$ norm with respect to input features of different dimensions become 1. $\alpha_\phi$ and $\beta_\phi$ share weights of neural network except for the last layer of MLP, so that we can get two quantities with single forward pass. This weight sharing method is employed in Levy et al. (2017) or other recent literature in learned optimization like in Metz et al. (2022b) and Metz et al. (2019).

Table 12: Results of ablation studies

| Dataset | L2E | ACC ↑ | NLL ↓ | ECE ↓ | KLD ↑ |
|---|---|---|---|---|---|
| Fashion-MNIST | Small L2E | 0.9107 | 0.2813 | 0.0217 | 0.1211 |
|  | Precond L2E | 0.9032 | 0.2922 | 0.0153 | 0.1322 |
|  | L2E | 0.9169 | 0.2451 | 0.0079 | 0.2465 |
| CIFAR-10 | Small L2E | 0.9062 | 0.2984 | 0.0486 | 0.6768 |
|  | Precond L2E | 0.8702 | 0.4082 | 0.0651 | 0.5189 |
|  | L2E | 0.9039 | 0.3074 | 0.0527 | 0.6481 |
| CIFAR-100 | Small L2E | 0.6866 | 1.1962 | 0.1498 | 1.4812 |
|  | Precond L2E | 0.5888 | 1.6447 | 0.2051 | 1.3674 |
|  | L2E | 0.6972 | 1.1346 | 0.1302 | 1.4659 |
| Tiny-ImageNet | Small L2E | 0.5360 | 2.0197 | 0.0795 | 2.4556 |
|  | Precond L2E | 0.3342 | 2.9357 | 0.1519 | 0.8814 |
|  | L2E | 0.5847 | 1.7339 | 0.1107 | 1.0747 |

### E.2 TASK DISTRIBUTION

**Dataset**  We use MNIST, Fashion-MNIST, EMNIST and MedMNIST for meta training. We do not use resized version of dataset for meta training. For MedMNIST, we use the BloodMNIST in the official website.

**Neural network architecture**  At each outer iteration, we randomly sample one configuration of neural network architecture which is constructed by the possible choice below. We use the following options for neural network architecture variation:

- Size of the convolution output channel : $\{4, 8, 16\}$
- Number of convolution layers: $\{1, 2, 3, 4, 5\}$
- Presence of the residual connection: boolean

### E.3 META-TRAINING PROCEDURE

**General hyperparameter**  For meta-training, we fix the length of the inner loop for all tasks at 3000 iterations. We determine this by monitoring the meta-objective during training and set sufficient length of inner loop for L2E to enter to high-density region regardless of tasks. This setting can vary when task distribution is changed. To compute the meta-objective, we collect 10 inner-parameters with a thinning interval of 50 during the last 500 iterations of the inner-loop. We find out that this thinning interval and the number of collected inner parameters do not have a significant impact on the model performance. Also, we use 1000 iterations of outer-loop for training meta-parameter since meta-loss converges after 1000 outer iterations.

**Outer optimization**  For training meta-parameter, we use Adam with learning rate 0.01 and $\beta_1 = 0.9, \beta_2 = 0.99$. We apply gradient clipping to the gradient of meta-objective to prevent unstable training due to the different gradient scale among tasks.

## F  ABLATION STUDIES

In Table 12, we demonstrate the results from our ablations studies. We evaluate how the size of task set can affect the generalization performance of L2E and the parameterization choice can make impact on the performance. We use following two variants of L2E

- **Small L2E**: meta-trained on only one dataset, using small architectures. In detail, this model only use Fashion-MNIST dataset and channel sizes of 4 and depths of 1 and 2 are possible choices of random configuration.
- **Precond L2E**: Parameterization of L2E changes from designing kinetic energy gradient to preconditioner.

Table 13: Hyperparameters for DE.

| Method | fashion-MNIST | CIFAR-10 | CIFAR-100 | Tiny-ImageNet |
|---|---|---|---|---|
| Optimizer | SGDM | SGDM | SGDM | SGDM |
| Num models | 100 | 100 | 100 | 10 |
| Total epochs | 5000 | 5000 | 5000 | 500 |
| Initial learning rate | 0.1 | 0.1 | 0.1 | 0.1 |
| Learning rate schedule | Cosine decay | Cosine decay | Cosine decay | Cosine decay |
| momentum decay | 0.5 | 0.1 | 0.1 | 0.1 |
| Weight decay | $5 \times 10^{-4}$ | $5 \times 10^{-4}$ | $1 \times 10^{-3}$ | $5 \times 10^{-4}$ |
| Batch size | 128 | 128 | 128 | 128 |

Table 14: Hyperparameters for CSGMCMC .

| Method | fashion-MNIST | CIFAR-10 | CIFAR-100 | Tiny-ImageNet |
|---|---|---|---|---|
| Num burn in epochs | 100 | 100 | 100 | 100 |
| Num models | 100 | 100 | 100 | 10 |
| Total epochs | 5100 | 5100 | 5100 | 600 |
| Thinning interval | 50 | 50 | 50 | 50 |
| exploration ratio | 0.8 | 0.8 | 0.94 | 0.8 |
| Step size | $2 \times 10^{-6}$ | $2 \times 10^{-6}$ | $2 \times 10^{-6}$ | $1 \times 10^{-6}$ |
| Step size schedule | Cosine | Cosine | Cosine | Cosine |
| Momentum decay | 0.5 | 0.5 | 0.1 | 0.1 |
| Weight decay | $5 \times 10^{-4}$ | $5 \times 10^{-4}$ | $1 \times 10^{-4}$ | $5 \times 10^{-4}$ |
| Batch size | 128 | 128 | 128 | 128 |

Firstly, we confirm that L2E trained with larger task distribution shows better generalization performance. Although small L2E shows decent performance on Fashion-MNIST and CIFAR-10, it shows significant drop of performance in Tiny-ImageNet and CIFAR-100. This implies that further scaling of task distribution can lead L2E to show better performance in diverse unseen tasks. Also, we compare the method parameterizing $D(z), Q(z)$ with neural networks as in Gong et al. (2018). This shows that our parameterization is significantly better than designing preconditioner. Since preconditioner matrix can only work as multiplier of learning rate or noise scale, it has limitation for its expressivity which shows limitation in BNNs.

## G   EXPERIMENTAL DETAILS

We used JAX library in our experiments.In order to facilitate reproducibility of our results, we provide not only the code and configurations. We use as an NVIDIA RTX-3090 GPU with 24GB VRAM and NVIDIA RTX A6000 with 48GB.

### G.1   REAL-WORLD IMAGE CLASSIFICATION

**Dataset**   We use tensorflow dataset for fashion-MNIST, CIFAR-10 and CIFAR-100. We utilize Tiny-ImageNet with image size of 64x64.

**Hyperparameter**   We report hyperparameters for each methods in Table 13, Table 14 and Table 16. We tune the hyperparameters of methods using BMA NLL as criterion with number of 10 samples. For all methods including L2E, we tune learning rate(step size), weight decay(prior variance) and momentum decay term. We use zero-mean gaussian as prior distribution for SGMCMC methods, so that prior variance is equal to half of the inverse of weight decay. For momentum decay, we grid search over $\alpha \in \{0.1, 0.3, 0.5\}$. For weight decay, we also search over $\lambda \in \{1e - 04, 5e - 04, 1e - 03\}$ to find best configuration. For step size, except for P-CSGMCMC, we search over $\epsilon \in \{1e - 06, 2e - 06, 1e - 05, 2e - 05, 4e - 05\}$. For P-CSGMCMC, due to preconditioning, we additionaly search over bigger step size including $1e - 04, 1e - 03$. In Table 17, we report the results for different values of hyperparameter of L2E. L2E shows robust performance with varying step size. Moreover, the performance of default value which is 0.5 shows satisfactory performance.

Table 15: Hyperparameters for P-CSGMCMC .

| Method | fashion-MNIST | CIFAR-10 | CIFAR-100 | Tiny-ImageNet |
|---|---|---|---|---|
| Num burn in epochs | 100 | 100 | 100 | 100 |
| Num models | 100 | 100 | 100 | 10 |
| Total epochs | 5100 | 5100 | 5100 | 600 |
| Thinning interval | 50 | 50 | 50 | 50 |
| exploration ratio | 0.8 | 0.8 | 0.8 | 0.8 |
| Step size | $2 \times 10^{-6}$ | $1 \times 10^{-3}$ | $1 \times 10^{-3}$ | $1 \times 10^{-4}$ |
| Step size schedule | Cosine | Cosine | Cosine | Cosine |
| Momentum decay | 0.1 | 0.1 | 0.1 | 0.1 |
| Weight decay | $5 \times 10^{-4}$ | $5 \times 10^{-4}$ | $1 \times 10^{-4}$ | $5 \times 10^{-4}$ |
| Batch size | 128 | 128 | 128 | 128 |

Table 16: Hyperparameters for L2E .

| Method | fashion-MNIST | CIFAR-10 | CIFAR-100 | Tiny-ImageNet |
|---|---|---|---|---|
| Num burn in epochs | 100 | 100 | 100 | 100 |
| Num models | 100 | 100 | 100 | 10 |
| Total epochs | 5100 | 5100 | 5100 | 600 |
| Thinning interval | 50 | 50 | 50 | 50 |
| Step size schedule | Constant | Constant | Constant | Constant |
| Momentum decay | 0.5 | 0.5 | 0.1 | 0.1 |
| Weight decay | $5 \times 10^{-4}$ | $5 \times 10^{-4}$ | $1 \times 10^{-3}$ | $5 \times 10^{-4}$ |
| Batch size | 128 | 128 | 128 | 128 |
| Step size | $2 \times 10^{-5}$ | $2 \times 10^{-5}$ | $1 \times 10^{-4}$ | $1 \times 10^{-5}$ |

Consequently, not only L2E requires fewer hyperparameters compared to existing methods, but it also requires less effort and cost for hyperparameter tuning.

**Baselines**    We use the following as baseline methods

- **Deep ensembles** (Lakshminarayanan et al., 2017) :  This method collects parameters trained from multiple different initialization for ensembling. DE is often compared with Bayesian methods in recent BNNs literature like in Izmailov et al. (2021b) in that DE induce similar function to HMC which is golden standard in BNNs with BMA.

- **Cyclical Stochastic Gradient MCMC** (Zhang et al., 2020): Zhang et al. (2020) introduced cyclic learning rate schedule to SGMCMC for improving exploration of sampler. CSGMCMC usually shows descent predictive performance comparing to other existing SGMCMC methods in large-scale experiments.

- **Preconditioned Cyclical Stochastic Gradient MCMC**: We additionaly apply Rmsprop style preconditioner to CSGMCMC.  We follow the implementation of Izmailov et al. (2021b).

**Metrics**    Let $p(y|x, \theta) \in [0, 1]^K$ be a predicted probabilities for a given input $x$ with label $y$ and $\theta$ is model parameter. $p^{(k)}$ denotes the $k$th element of the probability vector. We have the following common metrics on the dataset $\mathcal{D}$ consists of inputs $x$ and labels $y$:

- Accuracy (ACC):

$$\text{ACC}(\mathcal{D}) = \mathbb{E}_{(x,y)\in\mathcal{D}} \left[ \left[ y = \arg\max_k p^{(k)}(x) \right] \right]. \tag{13}$$

- Negative log-likelihood (NLL):

$$\text{NLL}(\mathcal{D}) = \mathbb{E}_{(x,y)\in\mathcal{D}} \left[ -\log p^{(y)}(x) \right]. \tag{14}$$

Table 17: L2E with varying step size.

| Dataset | Step size | ACC ↑ | NLL ↓ | ECE ↓ | KLD ↑ |
|---|---|---|---|---|---|
| fashion-MNIST | 1e-05 | 0.9140 | 0.2607 | 0.0129 | 0.2354 |
| | 2e-05 | 0.9169 | 0.2451 | 0.0079 | 0.2465 |
| CIFAR-10 | 1e-05 | 0.8965 | 0.3229 | 0.0498 | 0.8032 |
| | 2e-05 | 0.8968 | 0.3116 | 0.0520 | 0.6623 |
| | 4e-05 | 0.9039 | 0.3074 | 0.0527 | 0.6481 |
| CIFAR-100 | 1e-05 | 0.6824 | 1.1984 | 0.1230 | 2.4563 |
| | 2e-05 | 0.6889 | 1.1708 | 0.1252 | 2.5051 |
| | 4e-05 | 0.6986 | 1.1425 | 0.1277 | 2.1752 |
| Tiny-ImageNet | 1e-05 | 0.5847 | 1.7339 | 0.1107 | 1.0747 |
| | 2e-05 | 0.5801 | 1.7588 | 0.1239 | 0.8879 |

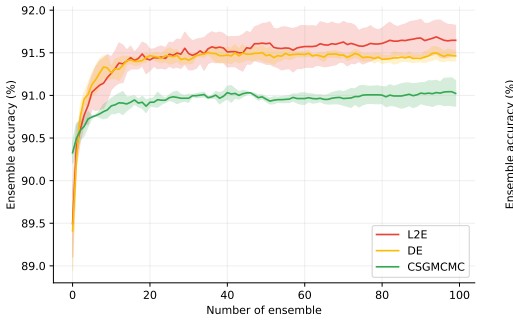 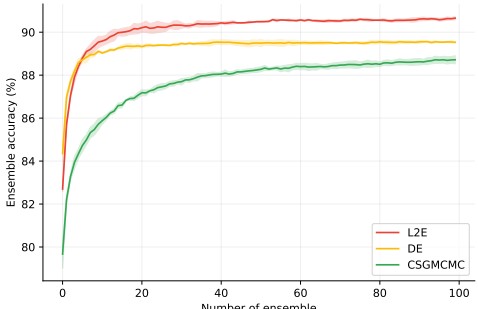

Figure 9: Ensemble accuracy as a function of number of ensemble, CNN on fashion-MNIST

Figure 10: Ensemble accuracy as a function of number of ensemble, ResNet20 on CIFAR-10

- Expected calibration error (ECE): Actual implementation of ECE includes dividing predicted probabilities with their confidence. We use following implementation

$$\text{ECE}(\mathcal{D}, N_{\text{bin}}) = \sum_{b=1}^{N_{\text{bin}}} \frac{n_b |\delta_b|}{n_1 + \cdots + n_{N_{\text{bin}}}}, \quad (15)$$

where $N_{\text{bin}}$ is the number of bins, $n_b$ is the number of examples in the $b$th bin, and $\delta_b$ is the calibration error of the $b$th bin. We use $N_{\text{bin}} = 15$ for computing ECE.

- Pairwise Kullback-Leibler Divergence(KLD): Given set of ensemble members $\Theta = \{\theta_i \dots \theta_M\}$, we construct matrix $A$ using pair of ensemble members which have $A_{ij} = p(y|x, \theta_i) \log \frac{p(y|x, \theta_i)}{p(y|x, \theta_j)}$ for a given input $x$ and $y$. We calculate the statistic as follows

$$D_{\text{KL}}(\mathcal{D}, M) = \mathbb{E}_{(x,y)\in\mathcal{D}} \left[ \frac{1}{M(M-1)} \sum_{i \neq j} A_{ij} \right]. \quad (16)$$

## G.2 ADDITIONAL VISUALIZATION

**Ensemble accuracy** We report the ensemble accuracy as a function of number of ensemble in other dataset setting in Figure 9, Figure 10 and Figure 11. They show that L2E can get the high performance with relatively small number of samples than DE and CSGMCMC. For this, we do not consider P-CSGMCMC as baseline since it shows significantly lower performance than other methods in predictive accuracy for large scale experiments.

**T-SNE visualization** We report the T-SNE visualization at each dataset in Figure 12 and Figure 13. Highlighted points represent parameters which are collected for BMA. L2E still shows specific pattern and covers almost the entire space to collect the samples in the function space.

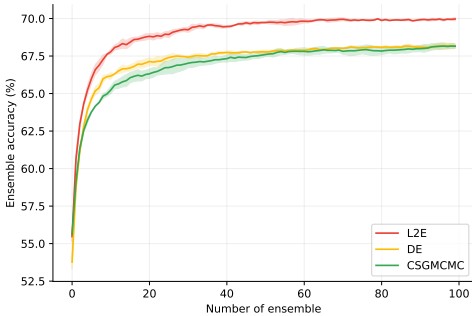

Figure 11: Ensemble accuracy as a function of number of ensemble, ResNet20 on CIFAR-100

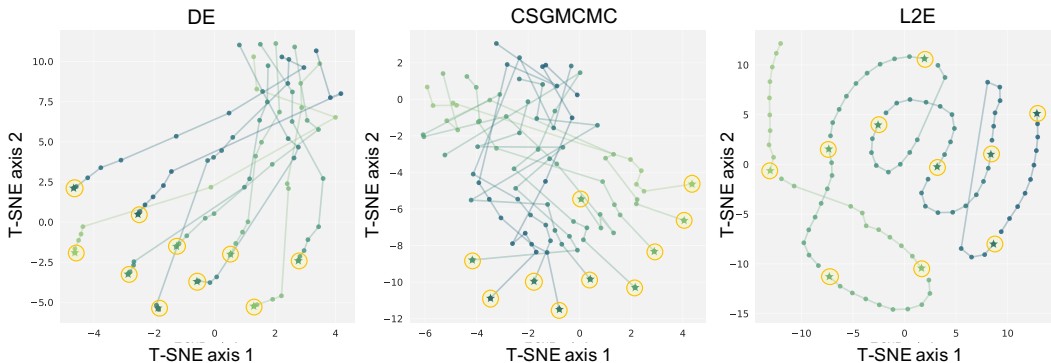

Figure 12: T-SNE visualization of learning trajectory. fashion-MNIST, CNN

**Loss surface**   We show the loss surface as a function of model parameters in a 2-dimensional subspace spanned by solutions of DE, CSGMCMC, P-CSGMCMC and L2E of each dataset in Figure 14, Figure 15 and Figure 16. L2E shows the most distinct modes except DE to collect the samples.

**Cosine similarity**   Figure 17 and Figure 18 show the heatmap of cosine similarity between weights of CNN on fashion-MNIST and ResNet20 on CIFAR-10. Still, DE and L2E collect more diverse solutions in weight space than CSGMCMC in other dataset.

**Training curve of L2E**   Figure 20 shows the training loss and nll of L2E in CIFAR-10 and CIFAR-100. L2E does not incur any spike in the loss value, meaning that it consistently search over high density region of posterior distribution.

### G.3   COMPUTATIONAL COST

Table 18 shows the actual computational time for each methods. L2E does not incur significant computational overhead. Consider L2E works well with extremely short thinning interval as in Figure 19, L2E is practically sound even with the additional computational cost.

### G.4   IMPLEMENTATION OF MCMC DIAGNOSTICS

We use Tensorflow Probability (Lao et al., 2020) library for calculating $\hat{R}$ and ESS. We use default parameter of the implementation. ESS is then normalized to the time consumed for running one thinning interval, which is equivalent to 50 epochs. Since ESS is too big to report since dimension of neural network parameters are huge for our experiments, we divide it by $10^{-5}$ and report for convenience.

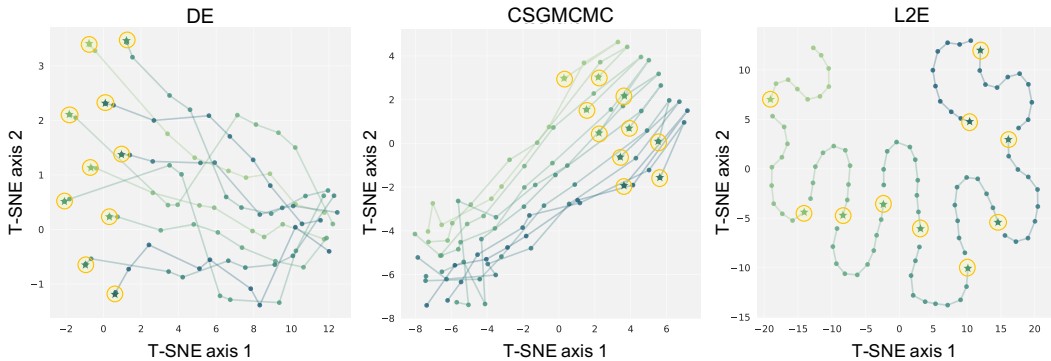

Figure 13: T-SNE visualization of learning trajectory. Tiny-ImageNet, ResNet56

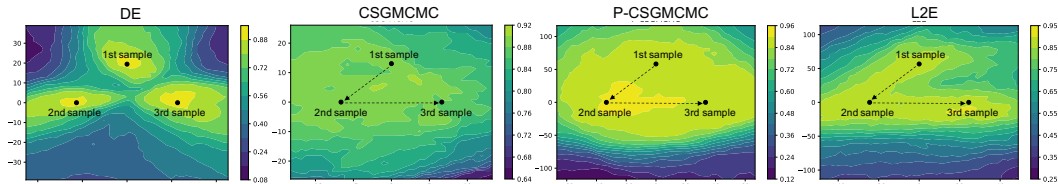

Figure 14: Loss surface of DE, CSGMCMC, P-CSGMCMC and L2E on fashion-MNIST, CNN

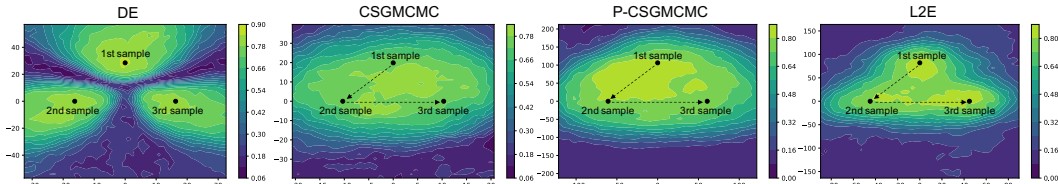

Figure 15: Loss surface of DE, CSGMCMC, P-CSGMCMC and L2E on CIFAR-10, ResNet20

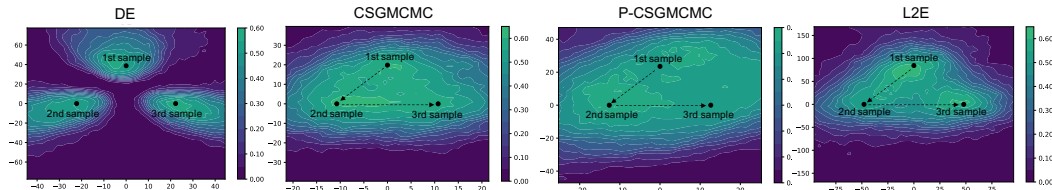

Figure 16: Loss surface of DE, CSGMCMC, P-CSGMCMC and L2E on CIFAR-100, ResNet20

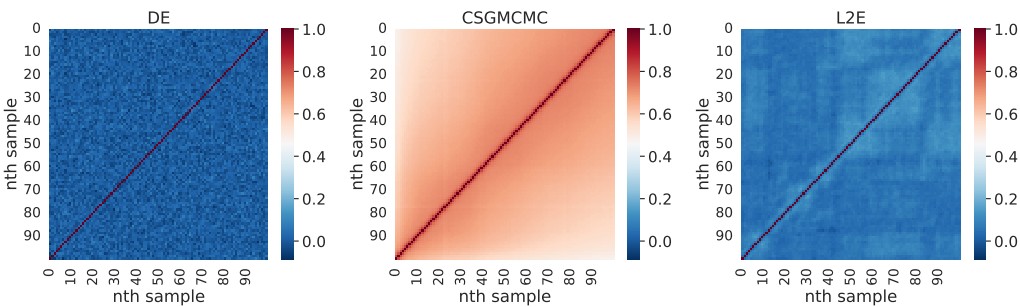

Figure 17: Cosine similarity between weights of CNN on fashion-MNIST.

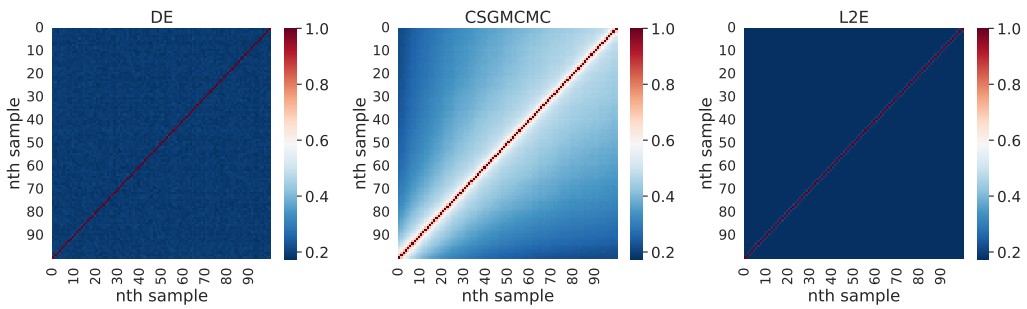

Figure 18: Cosine similarity between weights of ResNet20 on CIFAR-10.

Table 18: Wall clock time(sec) of SGMCMC methods per single epoch. Measured using NVIDIA RTX A6000.

| Method | fashion-MNIST | CIFAR-10 | CIFAR-100 | Tiny-ImageNet |
|---|---|---|---|---|
| CSGMCMC | 0.20 | 1.79 | 1.81 | 41.12 |
| P-CSGMCMC | 0.21 | 1.83 | 1.83 | 42.36 |
| L2E | 0.32 | 2.64 | 2.66 | 48.43 |

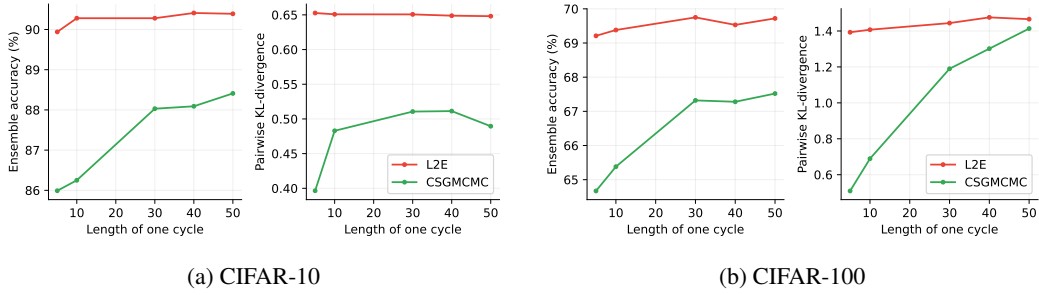

(a) CIFAR-10

(b) CIFAR-100

Figure 19: BMA accuracy and pairwise KLD by cycle length(thinning interval).

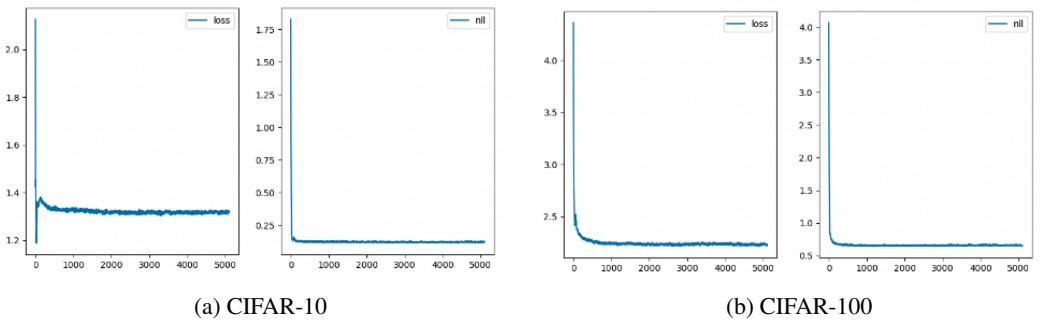

(a) CIFAR-10

(b) CIFAR-100

Figure 20: Training loss(NLL+ $l$2norm of parameters) and training mean NLL of L2E.

