# OpenReview forum: "Learning to Explore for Stochastic Gradient MCMC"
_ICLR.cc/2024/Conference — Submitted to ICLR 2024_

### Official Review · Reviewer_Stdr · 2023-10-12

**Soundness:** 2 fair
**Presentation:** 3 good
**Contribution:** 3 good
**Rating:** 5
**Confidence:** 4

**Summary:**

To encourage exploration, the authors proposed the "learning to explore approach" based on meta learning. The key idea is to learn the gradients of the kinetic energy  for SGMCMC update steps through two neural networks; the authors propose to train the networks on one tasks and then generalizes them to future tasks. This submission simplifies the work of meta-learning (Gong et al., 2018), which proposes to make D(z) and Q(z) as simple as possible.

**Strengths:**

The method starts from a practical viewpoint instead of physical intuitions to tackle the local trap problem suffered by the standard stochastic gradient MCMC methods. The idea of learning some key hyperparameters using parametrized networks based on meta learning may enjoy appealing implementation advantages.

The experimental evaluations, such as convergence analysis, loss surface demo, evaluation metrics (ACC, NLL, ECE), appear to be comprehensive in deep learning experiments.

**Weaknesses:**

(a) Underdamped Langevin/ Hamiltonian Monte Carlo is a good method for accelerating the overdamped alternative in terms of mixing rates. However, I believe it is far from explorative enough compared to other baseline methods, such as the replica-exchange based approaches [1,2]. I am not fully convinced if optimizing the kinetic energy is sufficient enough to solve the exploration problem. Your model input is $(\theta, r)$, I am not even sure it is really learning anything useful.

[1] Non-convex Learning via Replica Exchange Stochastic Gradient MCMC. ICML'20
[2] Non-reversible Parallel Tempering for Deep Posterior Approximation. AAAI'23.

(b) Empirically, your baselines such as deep ensemble and cyclical SGMCMC are pretty weak in terms of multi-modal simulations (although their optimization performance is acceptable).

**Questions:**

I am interested to see how your algorithm is compared to [2] or [3].

[2] Non-reversible Parallel Tempering for Deep Posterior Approximation. AAAI'23.

[3] Interacting Contour Stochastic Gradient Langevin Dynamics. ICLR'22

**************
*After rebuttal, I slightly increased my ratings from 3 to 5.*

---

> ### Author Response · Authors · 2023-11-17
>
> Thanks for constructive comments that can improve the quality of our paper. We would like to answer your comments.
>
> > Underdamped Langevin/ Hamiltonian Monte Carlo is a good method for accelerating the overdamped alternative in terms of mixing rates. However, I believe it is far from explorative enough compared to other baseline methods, such as the replica-exchange based approaches [1,2]. I am not fully convinced if optimizing the kinetic energy is sufficient enough to solve the exploration problem. Your model input is (θ,r), I am not even sure it is really learning anything useful.
>
> Firstly, we develop our method to enhance the exploration of single-chain MCMC from the meta-training stage. Considering the scale of modern machine learning, improving the exploration of single-chain methods is advantageous due to the significant memory costs associated with parallel chain-based approaches. Therefore, directly comparing the exploration of replica-exchange based methods that use multiple independent chains with our method seems unfair for our method. However, following your suggestion, we are now implementing the method proposed in [3]. We will upload the comparison with [3] as soon as experiments are completed.
>
> We have added the advantages of parameterizing kinetic energy term in overall response and Appendix A.1. Also, our learned sampler takes not only $(\theta,r)$ as inputs but also takes the gradient and moving average of the gradient which encode the posterior information as inputs. We explained details for inputs of neural networks in section 3.1 and Appendix E.1.
>
> > Empirically, your baselines such as deep ensemble and cyclical SGMCMC are pretty weak in terms of multi-modal simulations (although their optimization performance is acceptable).
>
> DE is known as one of the most effective ensemble methods, including Bayesian methods, in capturing both weight diversity and functional diversity [4]. While it is difficult to say DE is good at simulating BNNs posterior as it is a non-Bayesian method, it is a powerful approach at least in capturing multi-modality in the function and weight space. Also, CSGMCMC are currently widely used by practitioners as a baseline for efficient simulation of BNNs posterior with a single chain which is deeply related to our method. But, comparing with other methods will also be interesting, we will upload the comparison with [3] as soon as experiments are completed.
>
> [1] Non-convex Learning via Replica Exchange Stochastic Gradient MCMC. ICML'20
>
> [2] Non-reversible Parallel Tempering for Deep Posterior Approximation. AAAI'23.
>
> [3] Interacting Contour Stochastic Gradient Langevin Dynamics. ICLR'22
>
> [4] Ovadia, Yaniv, et al. "Can you trust your model's uncertainty? evaluating predictive uncertainty under dataset shift." Advances in neural information processing systems 32 (2019).

---

> > ### Author Response · Authors · 2023-11-22
> >
> > We comare our method with ICSGLD[1] in CIFAR-10 and CIFAR-100. For both methods, we do not use data augmentation and fix temperature hyperparameter to be 1. We collect 10 samples for both methods. We fix other hyperparameters same as in [1]. As shown in tables below , We confirm that L2E shows better predictive accuracy and predictive diversity than ICSGLD on CIFAR-10 and CIFAR-100. This results implies the strong exploration property of L2E shows even compared to parallel-chain based SGMCMC method.
> >
> > - CIFAR10 / ResNet-20
> > |Method | ACC | NLL | ECE | KLD|
> > |------ | :-----------: | :-----------: | :-----------: | :-----------:|
> > |ICSGLD | 0.883 ± 0.001 | 0.349 ± 0.002 | 0.079 ± 0.005 | 0.628 ± 0.087|
> > |ours(L2E) | 0.892 ± 0.003 | 0.334 ± 0.008 | 0.041 ± 0.002 | 0.639 ± 0.013|
> >
> > - CIFAR100 / ResNet-20
> > |Method | ACC | NLL | ECE | KLD|
> > |------ | :-----------: | :-----------: | :-----------: | :-----------:|
> > |ICSGLD | 0.655 ± 0.002 | 1.251 ± 0.005 | 0.087 ± 0.009 | 0.874 ± 0.011|
> > |ours(L2E) | 0.668 ± 0.013 | 1.211 ± 0.003 | 0.097 ± 0.002 | 1.412 ± 0.007|
> >
> >
> > [1] Interacting Contour Stochastic Gradient Langevin Dynamics. ICLR'22
> >
> >
> >
> > Thank you for your time in reviewing our work. Have we addressed your concerns raised in the initial review? The author discussion period will end tomorrow, so we would like to take the remaining time to ensure we have addressed any remaining concerns you may have. Please let us know if anything needs further clarification.
> > Thank you, Authors

---

> ### Comment · Reviewer_Stdr · 2023-11-22
> **Thank for discussing more exploration-related baselines**
>
> **Deep Ensemble is a non-Bayesian method**
>
> I may respectfully disagree with this (although some debate exists) and I believe deep ensemble is also a Bayesian method. First, SGD is an approximate Bayesian method to capture one major mode [1]; second, deep ensemble uses different initialization to capture different modes, which is why I believe that the **exploration solely driven by different initialization may not be sufficient to tackle explorations**. Nevertheless, the straightforward advantage is that it is easy to implement, which is why it is so popular. But fundamentally it is not a principled method.
>
> [1] Stochastic Gradient Descent as Approximate Bayesian Inference. JMLR'17.
>
> **a single chain that is deeply related to our method**
>
> I understand why the authors didn't compare the multiple-chain baselines. However, in my experience, they are really crucial to address the exploration problem. In my limited experience in motivating explorations, how to appropriately set up different **learning rates** and **temperatures** is crucial (in theory and also well supported in practice) and more important than the rest of the factors, such as momentum, etc. So these references on simulated tempering (one chain version) and parallel tempering (multi-chain version) cannot be missing in any papers that aim to facilitate exploration.
>
>
> **Popularity is not the unique criterion as a baseline.**
>
> Many popular Github repos also work quite well, but they are not published as papers. **Popularity is not the first principle** to choose baselines in academic papers.
>
> **Summary**
>
> I will slightly increase my rating from 3 to 5 to thank the authors including one more baseline.
>
> I would suggest the authors include more relevant experiments to justify your work in your next revision. For example, instead of running so many large-scale experiments, I am actually more interested to see how your algorithm performs on the 25-mode 2D mixture distribution with unequal weights (figure 3 in [1]). If your algorithm can achieve similar performance without introducing too much cost, I will be more convinced if the algorithm is good or not. (I don't expect your algorithm to outperform replica-exchange.)
>
> [1] INTERACTING CONTOUR STOCHASTIC GRADIENT LANGEVIN DYNAMICS

---

### Official Review · Reviewer_36ng · 2023-10-23

**Soundness:** 3 good
**Presentation:** 2 fair
**Contribution:** 2 fair
**Rating:** 5
**Confidence:** 5

**Summary:**

Unlike prior work, the authors propose a meta-learning strategy for SGHMC by learning the kinetic energy term. The authors show that the meta-learning the kinetic energy term is able to generalize as well as other competing methods, beyond the data distributions during meta-training. Such an approach leads to improved OOD detection, and the authors experimentally demonstrate the discovery of disconnected modes.

**Strengths:**

- The authors take a different perspective to meta-learned SGMCMC where instead of the prior work which focused on meta-learning the dynamics matrix $D$ and curl matrix $Q$, they meta-learn the kinetic energy term.
- The meta-learning procedure is designed to be operationally fairly simple, and can be relatively easily accommodated into existing SGMCMC pipelines.

**Weaknesses:**

- I think the first and the biggest weakness of the work lies in the way it is framed. It seems that you are trying to compete with deep ensembles, and other SGMCMC variants. In fact, the message I get from the experimental results is that L2E is that it performs pretty much worse is most cases and at worse computational cost. But one could argue, the promise of the method is rather in the generalizability of the meta-learning procedure to unseen datasets. In a sense, it is pretraining for SGMCMC, which to me is the most interesting. Unfortunately, this is not how the authors position this paper.
- The deliberate choice to not use data augmentations is a little concerning. I understand that it does not neatly fit into the Bayesian perspective, but when the same datasets perform significantly better with data augmentation (including with SGMCMC), it makes the comparisons incomplete. It would not be too hard to add data augmentations as is into the same training setup, and make the comparison on these datasets more fair to modern deep learning that achieves both better accuracy and better calibration properties.
- The comparisons to DE and cSGMCMC looks to me a little unfair. L2E sees more variety of data by design, and one could then posit that the better OOD detection is not that surprising and simply a matter of having seen broader distributions are meta-train time.
- It would be really helpful if we could have a toy illustration to build intuitions. For instance, meta-learning datasets sampled from 1-D sines and cosines. It would also reveal how the method behaves with "in-between" uncertainty, i.e. parts of the input space not in the training dataset. It would also help to see how the sampler behaves beyond the training dataset, when say you change to say a different function family of polynomials.

### Minor

- The choice of citation in the last paragraph on Page 1 "Zhang et. al., 2020" alone is odd. I would recommend citing all the other works.
- The authors have made the choice of not having an explicit related work section. I think that is alright, but other work must be contextualized somewhere in the main text. Part of it is done in

**Questions:**

1. Could the authors clarify how L2E works at test time? Do you run the `InnerLoop` and take the samples after appropriate steps, thinning, and the compute the BMA?
2. Equation 8 states that the loss is computed on validation data point. Is that true? Or during training this comes from the set of training distributions.
3. The number of epochs as reported in Table 10 seem incredibly high for such a method to be scalable at all. Or is this supposed to be the number of gradient steps?
4. How are the parameters initialized in the outer loop? Do you rely on default initializations for the family of architectures considered for meta-training? I wonder if such an approach would be unstable if the nature of networks considered is different.
5. It seems like the choice to have different architectures for meta-training is challenging. What happens when the images have different number of channels? How is that currently handled? Are different sizes of images handled via resizing?
6. I may have missed it, but I did not find the number of inner loop steps. Is it supposed to be the number of epochs?
7. At test time, is the number of inner loop steps the same as train time? What happens when it is lesser? What happens when you keep it much larger?

---

> ### Author Response · Authors · 2023-11-17
>
> Thanks for constructive comments that can improve the quality of our paper. We would like to answer your comments.
>
> > I think the first and the biggest weakness of the work lies in the way it is framed. It seems that you are trying to compete with deep ensembles, and other SGMCMC variants. In fact, the message I get from the experimental results is that L2E is that it performs pretty much worse is most cases and at worse computational cost. But one could argue, the promise of the method is rather in the generalizability of the meta-learning procedure to unseen datasets. In a sense, it is pretraining for SGMCMC, which to me is the most interesting. Unfortunately, this is not how the authors position this paper.
>
> Would you elaborate why L2E performs pretty much worse than other methods? According to Table 1, we outperform other baselines in general. Only DE can match the predictive accuracy in some experiments and also L2E shows the best performance in OOD detection. Except for distribution-shift experiment where Bayesian methods can often fail [1], L2E shows competitive experiments with DE and outperforms CSGMCMC. From a computational cost perspective, we only meta-train once and L2E only takes about 1.2~1.5 times more wall clock time for one step update than CSGMCMC according to Table 18. Also, L2E is potentially much more efficient than other baselines since it is robust to the choice of thinning interval. In Figure 19, we plot the performance of the sampler with varying cycle length (thinning interval). Although L2E uses more computational cost for one step update, L2E can achieve comparable accuracy with other baselines with smaller total computational cost. We believe L2E excels in both aspects (performance and generalization), as it achieves better performance with reasonable computation costs even on unseen datasets compared to other methods.
>
> [1] Izmailov, Pavel, et al. "Dangers of Bayesian model averaging under covariate shift." Advances in Neural Information Processing Systems 34 (2021): 3309-3322.
> .

---

> > ### Author Response · Authors · 2023-11-17
> >
> > > The deliberate choice to not use data augmentations is a little concerning. I understand that it does not neatly fit into the Bayesian perspective, but when the same datasets perform significantly better with data augmentation (including with SGMCMC), it makes the comparisons incomplete. Make the comparison on these datasets more fair to modern deep learning that achieves both better accuracy and better calibration properties.
> >
> > Since including data augmentation violates i.i.d assumption of the dataset which is commonly assumed for Bayesian methods, this can lead to model misspecification problem. Therefore, prior work such as [1] argued the incompatibility between Bayesian methods and data augmentation. However, in modern machine learning, data augmentation is an indispensable technique. Thus, in response to your request, we conduct experiments with data augmentation on CIFAR-10 and CIFAR-100. Please refer to Appendix C.2 for full details and results of the experiments. Since using data augmentation introduces cold posterior effect [2] for Bayesian methods such as CSGMCMC and L2E, we additionally tune the temperature hyperparameter for these methods.
> >
> > According to Tables, in terms of predictive accuracy, DE outperforms other baselines on CIFAR-10 and L2E beats other baselines on CIFAR-100. With data augmentation, it is not very surprising that DE outperforms other Bayesian methods like L2E and CSGMCMC in terms of predictive accuracy since they suffer from model misspecification and temperature tuning can partially handle this problem [3]. This can explain why L2E shows the worst NLL in CIFAR-10 experiment since likelihood does not fit with the observation. When it comes to predictive diversity, L2E significantly outperforms baselines on both experiments. Although applying data augmentation introduces significant variations to the posterior landscape, we confirm that L2E still maintains the exploration property. To sum up, we argue that L2E is still a practical method even with data augmentation since it shows competitive predictive performance and efficiently explores the posterior landscape.
> >
> > [1] Izmailov, Pavel, et al. "What are Bayesian neural network posteriors really like?." International conference on machine learning. PMLR, 2021.
> >
> > [2]  Wenzel, Florian, et al. "How good is the bayes posterior in deep neural networks really?."
> >
> > [3] Kapoor, Sanyam, et al. "On uncertainty, tempering, and data augmentation in bayesian classification." Advances in Neural Information Processing Systems 35 (2022): 18211-18225
> >
> > - CIFAR10 / ResNet-20
> > |Method | ACC | NLL | ECE | KLD|
> > |------ | :-----------: | :-----------: | :-----------: | :-----------:|
> > |DE | 0.927 ± 0.001 | 0.218 ± 0.002 | 0.021 ± 0.001 | 0.238 ± 0.001|
> > |CSGMCMC | 0.923 ± 0.000 | 0.239 ± 0.003 | 0.014 ± 0.000 | 0.125 ± 0.001|
> > |ours(L2E) | 0.924 ± 0.003 | 0.249 ± 0.001 | 0.042 ± 0.002 | 0.332 ± 0.003|
> >
> > - CIFAR100 / ResNet-20
> > |Method | ACC | NLL | ECE | KLD|
> > |------ | :-----------: | :-----------: | :-----------: | :-----------:|
> > |DE | 0.707 ± 0.001 | 1.053 ± 0.002 | 0.096 ± 0.005 | 0.515 ± 0.002|
> > |CSGMCMC | 0.697 ± 0.001 | 1.066 ± 0.007 | 0.064 ± 0.001 | 0.373 ± 0.005|
> > |ours(L2E) | 0.710 ± 0.001 | 1.053 ± 0.005 | 0.053 ± 0.001 | 0.911 ± 0.007|

---

> > > ### Author Response · Authors · 2023-11-17
> > >
> > > > The comparisons to DE and cSGMCMC looks to me a little unfair. L2E sees more variety of data by design, and one could then posit that the better OOD detection is not that surprising and simply a matter of having seen broader distributions are meta-train time.
> > >
> > > DE is currently a widely referenced method for OOD detection [1, 2] and it is not that strange to compare our approach with DE and CSGMCMC even though L2E was trained on additional datasets during the meta-training. We believe this comparison is realistic and fair because we did not explicitly meta-train L2E for OOD detection during meta-training. Furthermore, we conduct additional ablation experiments to assess whether seeing more datasets during meta-training affects the performance of OOD detection. In the following Table, we confirm that seeing more datasets during the meta-training stage do not necessarily lead to better OOD detection performance. Small L2E which is meta-trained only on Fashion-MNIST shows almost similar OOD detection performance with L2E or even better performance for some datasets. We think that meta-training on external datasets can rather lead to overconfidence of model on meta-training datasets which can potentially decrease OOD detection performance.  Also, we have added 1-D synthetic regression task in Appendix C.1. and L2E also shows competitive performance at capturing epistemic uncertainty in regression task. Since 1-D synthetic dataset is completely different from the meta-training datasets, it is difficult to say that L2E shows good OOD detection performance due to the external sources of the dataset.
> > >
> > > |In-dist|OOD|Small L2E|L2E
> > > |:---:|:---:|:---:|:---:|
> > > |CIFAR10|CIFAR100|**0.855±0.001**|0.851±0.001|
> > > |   |SVHN|0.929±0.001|**0.931±0.003**|
> > > |   |Tiny-ImageNet|**0.855±0.001**|0.853±0.001|
> > > |CIFAR100|CIFAR10|0.789±0.003|**0.792±0.002**|
> > > |   |SVHN|0.912±0.004|**0.917±0.003**|
> > > |   |Tiny-ImageNet|0.777±0.001|**0.782±0.003**|
> > >
> > >
> > > [1] Lakshminarayanan, Balaji, Alexander Pritzel, and Charles Blundell. "Simple and scalable predictive uncertainty estimation using deep ensembles." Advances in neural information processing systems 30 (2017).
> > >
> > > [2] Ovadia, Yaniv, et al. "Can you trust your model's uncertainty? evaluating predictive uncertainty under dataset shift." Advances in neural information processing systems 32 (2019).
> > >
> > > > It would be really helpful if we could have a toy illustration to build intuitions. For instance, meta-learning datasets sampled from 1-D sines and cosines. It would also reveal how the method behaves with "in-between" uncertainty, i.e. parts of the input space not in the training dataset. It would also help to see how the sampler behaves beyond the training dataset, when say you change to say a different function family of polynomials.
> > >
> > > To visually check whether L2E can capture the epistemic uncertainty, we have added 1-D synthetic regression experiment in Appendix C.1. We use DE and HMC as baseline methods. Please refer to Appendix C.1 for experimental setup and Figure 7 for the regression results. We find that even without meta-training on synthetic 1-D dataset generated from particular polynomials, our method captures ‘in-between’ uncertainty well. As shown in Figure 7, L2E produces diverse predictions for out-of-distribution data points. HMC shows the best performance in capturing ‘in-between’ uncertainty which aligns with results from other works such as [1], but we believe that L2E still has sufficient capabilities to capture ‘in-between’ uncertainty in various tasks.
> > >
> > > [1] D'Angelo, Francesco, Vincent Fortuin, and Florian Wenzel. "On stein variational neural network ensembles." arXiv preprint arXiv:2106.10760 (2021).
> > >
> > > > The choice of citation in the last paragraph on Page 1 "Zhang et. al., 2020" alone is odd. I would recommend citing all the other works.
> > >
> > > Thanks for pointing this out. We have added other works in the last paragraph on Page 1.
> > >
> > > > The authors have made the choice of not having an explicit related work section. I think that is alright, but other work must be contextualized somewhere in the main text. Part of it is done in
> > >
> > > Thanks for pointing this out. We have added related work section in the Appendix D.

---

> > > > ### Author Response · Authors · 2023-11-17
> > > >
> > > > > Could the authors clarify how L2E works at test time? Do you run the InnerLoop and take the samples after appropriate steps, thinning, and the compute the BMA?
> > > >
> > > > Yes we run InnerLoop (Algorithm 2 in the paper) trained meta-parameter. It is essentially identical to other SGMCMC methods during test time.
> > > >
> > > > > Equation 8 states that the loss is computed on validation data point. Is that true? Or during training this comes from the set of training distributions.
> > > >
> > > > We split the Datasets used for the meta-training into training, validation, and test sets. We run InnerLoop using the training split to collect parameters, and BMA meta-loss is calculated using the data sampled from the validation split.
> > > >
> > > > > The number of epochs as reported in Table 10 seem incredibly high for such a method to be scalable at all. Or is this supposed to be the number of gradient steps?
> > > >
> > > > Due to the multi-modal and highly complex nature of the loss surface for large-scale BNNs posteriors, it is necessary to run many epochs to allow samplers to explore various parts of the posterior. We believe that collecting a diverse set of parameters and performing Bayesian Model Averaging (BMA) is important for BNNs. You can easily find other papers like [1]  that use more than 10000 epochs for collecting samples in CIFAR-10.
> > > >
> > > > [1] Izmailov, Pavel, et al. "What are Bayesian neural network posteriors really like?." International conference on machine learning. PMLR, 2021.
> > > >
> > > > > How are the parameters initialized in the outer loop? Do you rely on default initializations for the family of architectures considered for meta-training? I wonder if such an approach would be unstable if the nature of networks considered is different.
> > > >
> > > > We use default initialization for neural network architectures in meta-training tasks. This may cause instability during meta-training, but we did not experience that problem during meta-training. If this happens, splitting the task distribution into groups of similar neural network architectures and dividing meta-training into different phases as in [1] can be one solution.
> > > >
> > > > [1] Metz, Luke, et al. "Velo: Training versatile learned optimizers by scaling up." arXiv preprint arXiv:2211.09760 (2022).
> > > >
> > > > > It seems like the choice to have different architectures for meta-training is challenging. What happens when the images have different number of channels? How is that currently handled? Are different sizes of images handled via resizing?
> > > >
> > > > It is true that sampling various architectures and datasets for meta-training is more challenging than training on a single task. The task distribution we propose in the paper includes datasets with varying numbers of channels. However, the crucial point is that, even if the number of channels or dimensions in the input changes, the meta-learning process itself does not fundamentally change. Since the neural network is initialized based on the shape of the sampled dataset when the task is sampled from task distribution, we do not handle it in a different way than in single-task learning.
> > > >
> > > > >I may have missed it, but I did not find the number of inner loop steps. Is it supposed to be the number of epochs?
> > > >
> > > > We use 3000 steps of inner-loop for each outer-iteration. This is not supposed to be the number of epochs, but it is determined to be a sufficient number of steps that sampler requires to converge on meta-training tasks. We will make this clearer in the revised version.
> > > >
> > > > > At test time, is the number of inner loop steps the same as train time? What happens when it is lesser? What happens when you keep it much larger?
> > > >
> > > > If we use a well-trained sampler like L2E, the number of steps for running the sampler during test time is not related to the inner loop steps used at train time. Inner loop steps are set to train the sampler on relatively simple meta-training tasks.  We already use much larger inner loop steps at test time than training time since we run L2E for 5100 epochs in CIFAR-10 experiment, which is more than 2 million update steps. But using too short innerloop steps in the meta-training stage interrupts the sampler to learn the geometry of various parts of the loss surface during the meta-training so this can cause divergence of sampler at test-time. This phenomenon has also been addressed in the papers like [1] and [2] in the literature of learned optimization.  Also, we have added ablation study for the impact of the number of inner loop steps in Appendix C.3.
> > > >
> > > > [1] Metz, Luke, et al. "Understanding and correcting pathologies in the training of learned optimizers." International Conference on Machine Learning. PMLR, 2019.
> > > >
> > > > [2] Metz, Luke, et al. "Velo: Training versatile learned optimizers by scaling up." arXiv preprint arXiv:2211.09760 (2022).

---

> > > ### Comment · Reviewer_36ng · 2023-11-17
> > > **Response to authors**
> > >
> > > > Thus, in response to your request, we conduct experiments with data augmentation on CIFAR-10 and CIFAR-100. Please refer to Appendix C.2 for full details and results of the experiments
> > >
> > > I really appreciate you running these experiments. Here again, I want to point out that a performance less than 93% on CIFAR-10 seems like a model that is not up to par. It makes the comparisons look weak. But in any case, as you see, the difference in the methods become far less stark.
> > >
> > > We as a community of Bayesian deep learning have been underselling these results for years. Without strong results that at least match simple baselines in deep learning, I am afraid BDL remains an intellectual curiosity with nothing exciting for the broader deep learning community to appreciate.

---

> > ### Comment · Reviewer_36ng · 2023-11-17
> > **Response to authors**
> >
> > > According to Table 1, we outperform other baselines in general.
> >
> > Perhaps, you and I have different expectations from what Bayesian methods should contribute on top of deep learning. Let me present my point of view. Table 1, by the choice of not using data augmentations, already seems to perform below par what a single neural network can provide. Therefore, by having a weaker baseline to compare to, I think it is unfair to claim "outperform". Next, perhaps one place where I would expect Bayesian methods to really shine (other than accuracy which we now know that Bayesian methods can help) would be the quality of uncertainty. For a moment, the ECEs are still fairly far off from the best performing P-cSGMCMC. If as a user of the method, I am spending all the extra compute to construct posterior samples, I would ideally like to be at least very close other baselines even if not outperform. The gaps are fairly large, e.g. on Tiny ImageNet as large as 4% v/s 11% for ECE, and multiple percentage points on others.
> >
> > Table 3 for OOD detection tells me, that as the datasets get harder, the benefit of the method seems to become less remarkable. For a time, when scaling of models is important for performance, such a method seems not so amenable to scaling with the datasets.
> >
> > But again, I don't want to force you into the narrative that L2E should be the absolute best. That is certainly the wrong goal to strive for. Nevertheless, outperforming would be a strong characterization of the results, and would require further work to really make that claim.
> >
> > > From a computational cost perspective, we only meta-train once and L2E only takes about 1.2~1.5 times more wall clock time for one step update than CSGMCMC according to Table 18
> >
> > I think that is completely reasonable. A large fraction of deep learning uses pre-trained models, so we can potentially even ignore this cost for the sake of our discussion. And as you say, with sufficient meta-training, it might even be effectively lower. I am indeed very happy to see that the meta-learned samplers are not bad at all, and this is not a big concern for me.

---

> ### Comment · Reviewer_36ng · 2023-11-17
> **Response to authors**
>
> > we confirm that seeing more datasets during the meta-training stage do not necessarily lead to better OOD detection performance.
>
> Thank you for conducting this experiment.
>
> I have a different reading of the results here. On CIFAR-100, a broader dataset, large version seems to win always. Of course, it would be hard to extrapolate, but it would appear that a more diverse dataset leads to better sampler and better uncertainty estimation. Wouldn't that be the ideal scenario for L2E anyways - it improves its uncertainty estimation with more diverse meta-training?
>
> In any case, what I was hinting at perhaps not a criticism of the approach, but more of trying to understand where the benefits come from - parametrizing the kinetic energy term or the choice of diverse datasets. In the current work, both these choices haven't quite been studied independently.

---

> > ### Comment · Reviewer_36ng · 2023-11-17
> > **Response to authors**
> >
> > I appreciate the effort the authors have taken to add all the new experiments.
> >
> > I still think that there is a strong push to convey the method as an "outperformer" over existing methods, but I think the baselines are weak, and the attribution of method's benefits remain entangled between the choice of kinetic energy parametrization and the choice of using diverse meta-training. Further, the baselines used are below par for the deep learning community. We cannot rely on the intellectual merits of Bayesian inference alone, the benefits need to show up somewhere either in the accuracy or uncertainty properties, which I don't think is convincing when reading between the lines of what the results tables portray (as I have outlined in my comments above for additional experiments). Therefore, I will keep my score.
> >
> > This perhaps could have been a different paper, were it not concerned with outperforming and introducing the idea that meta-training and kinetic energy parametrization is something that the community has largely ignored. But of course, it is the authors' right to decide, and I want to make sure the authors know that I do not hold it against them in my assessment.

---

> > > ### Author Response · Authors · 2023-11-22
> > >
> > > Thanks for your response. We summarize your concerns in three categories and address your concerns.
> > >
> > > 1. Weak predictive performance of baselines and L2E:
> > >
> > > In the experiments of our paper, we use 50 epochs for each solution of DE and use a thinning interval of 50 epochs for CSGMCMC and L2E. This experimental setup is designed to fairly evaluate all methods using similar training budget. Without data augmentation, this training budget is sufficient for all methods to achieve the best performance. Therefore, we do not intentionally make the baselines weak in our setup. However, when data augmentation is applied, we observe difficulties in achieving the highest performance with only 50 epochs especially for DE. Without limiting the computational cost, DE shows almost 93.5% of accuracy in CIFAR-10. However, in this case, DE uses approximately 4 times more number of epochs compared to L2E. Therefore, considering computational cost, we assert that L2E is sufficiently comparable to DE and CSGMCMC.
> > >
> > > 2. Weak uncertainty estimation performance of L2E:
> > >
> > > Although P-CSGMCMC shows strong results in terms of uncertainty quantification, this method shows inferior performance with respect to predictive accuracy. It is reasonable to compare uncertainty quantification between methods with similar predictive performance. Except for P-CSGMCMC, L2E consistently demonstrates comparable ECE to other methods. Furthermore, our additional synthetic regression experiment in Appendix C.1. highlights the strength of L2E in uncertainty estimation.
> > >
> > > 3. Effect of task diversity on uncertainty estimation and OOD detection:
> > >
> > > According to Table 10, when models are trained on CIFAR-10, small L2E(L2E trained on smaller tasks) shows better OOD detection performance on Tiny ImageNet, which implies that better OOD detection performance on a broader dataset does not necessarily come from task diversity. Through the ablation studies in Table 12, we examine the relationship between uncertainty estimation and task diversity in image classification experiments. While L2E shows better predictive performance than other variants, it does not show better ECE than others. Unfortunately, it is difficult to claim that diversifying meta-training tasks leads to better uncertainty estimation and OOD detection performance. Also, we compare L2E with Precond L2E (L2E parameterized with the learnable preconditioner) in Table12, showing that kinetic energy parameterization is better than parameterizing preconditioner in terms of predictive accuracy and uncertainty estimation.
> > >
> > > We appreciate your recognition of the potential of our paper. However, we have already demonstrated, through ablation studies, that kinetic energy parameterization and diverse meta-training tasks contribute to predictive accuracy and generalization capacity, respectively. While the overall performance may be lower than what the deep learning community expects, we compare our method with strong baselines under fair experimental setup. Thank you for providing insights to further enhance our research.

---

### Official Review · Reviewer_LiEv · 2023-10-29

**Soundness:** 2 fair
**Presentation:** 1 poor
**Contribution:** 2 fair
**Rating:** 3
**Confidence:** 4

**Summary:**

This paper introduces a meta-learning method to learn the kinetic energy term in SGMCMC for multi-modal distributions. The proposed method uses two neural networks (NNs) to parameterize kinetic energy term and train these NNs based on a meta-objective function, which is the validation loss. The authors have conducted several experiments, including image classification and out-of-distribution detection to demonstrate the proposed method.

**Strengths:**

1.	The methodology of the proposed method is simple, which makes it a practical method for many tasks.
2.	The experiments and ablation studies are comprehensive and cover many aspects of sampling, including predictive accuracy, uncertainty quantification, convergence diagnostic

**Weaknesses:**

1.	The paper did not mention at all at the beginning that, there already exist meta-learning methods for SGMCMC, such as Gong et al. Only in Section 3.1, the authors first briefly mention that paper. The presentation is misleading and may give the impression that this paper is the first to study meta-learning for SGMCMC.
2.	More importantly, the proposed method is essentially very similar to Gong et al, which uses the same formulation for the SGMCMC class, but will slightly different parameterization. Gong et al learns the curl matrix Q and the diffusion matrix D whereas the proposed method learns the kinetic energy term. Given the similarity, it is important to clearly state the difference compared with Gong et al.
3.	The motivation of the proposed method is weak. Since the main difference compared with Gong et al is the parameterization of the kinetic energy term, it is important to clearly motivate this choice. The authors did not mention at all in the paper why they choose to learn the kinetic energy term rather than the curl matrix and the diffusion matrix. The advantages of doing so are not discussed.
4.	Gong et al has proposed several meta objectives. Again, the authors did not discuss their meta objective with existing ones.
5.	In experiments, the authors did not compare with Gong et al, which is a very related method.
6.	Although the experimental results of the proposed method are promising, it is not clear why the proposed meta-learning method can lead to these empirical improvements. What kind of learning updates did the meta-learning algorithm learn in the end?  Why does the method show a better exploration-exploitation balance? how does it achieve that?
7.	The meta-learning tasks are classification on MNIST, Fashion-MNIST, EMNIST and MedMNIST and the downstream tasks are classification on Fashion-MNIST, CIFAR and Tiny ImageNet.  Since Fashion-MNIST has appeared in meta-learning tasks, is it reasonable to use it as a test task? Are CIFAR and Tiny ImageNet similar to MNIST to be considered as test tasks?

Gong et al, Meta-Learning For Stochastic Gradient MCMC, ICLR 2019

**Questions:**

1.	Can you explain T-SNE visualization of learning trajectories? Why does L2E’s trajectory look very different from DE and CSGMCMC?

---

> ### Author Response · Authors · 2023-11-17
>
> Thanks for constructive comments that can improve the quality of our paper. We would like to answer your comments. Also, we revise our paper with comprehensive comparison with Gong et al., in Appendix A. Please refer to  Appendix A for more detailed information.
>
> > The paper did not mention at all at the beginning that, there already exist meta-learning methods for SGMCMC, such as Gong et al. Only in Section 3.1, the authors first briefly mention that paper. The presentation is misleading and may give the impression that this paper is the first to study meta-learning for SGMCMC.
>
> Sorry for not faithfully discussing Gong et al., in our main paper, we did not intend to be misleading in acknowledging its contribution. We could not put enough discussion in the main paper due to the page limit. In the revised version, we added the discussion about Gong et al., and devoted an additional section in the appendix to compare ours to Gong et al., in various aspects.
>
> > More importantly, the proposed method is essentially very similar to Gong et al, which uses the same formulation for the SGMCMC class, but will slightly different parameterization. Gong et al learns the curl matrix Q and the diffusion matrix D whereas the proposed method learns the kinetic energy term. Given the similarity, it is important to clearly state the difference compared with Gong et al.
>
> We have added Table 6 and Appendix A to state the difference between Gong et al. Please refer to Appendix A and Table 6 for more detailed explanation.
>
> For what you have pointed out, especially the statement that ours is very similar to Gong et al., because they share the same formulation for the SGMCMC class and differ only in parameterization. We respectfully disagree with this. The SGMCMC class is a complete class, so any convergent SGMCMC algorithm must belong to this class, so whatever SGMCMC algorithm we try to learn we always have to be in the class both ours and Gong et al., are following. Moreover, as we have detailed in Appendix A, the fact that we are learning kinetic energy is a significant innovation, bringing noticeable gains both in terms of the exploration behaviour of the sampler and efficient training.
>
> Moreover, we have different goals with Gong et al. Their purpose was to learn the sampler that fastly converges to high density region with low bias. However, our approach is specifically designed with more concrete purpose of effectively simulating multi-modal BNNs posterior distribution and also generalizing to unseen problems for practicality. This goal was not achieved by Gong et al because Figure 3 in Gong et al showed that learned sampler fastly converges to low energy region, but learned friction term $D_f$ restricts the amount of update in low energy region to prevent divergence which harms the exploration of the sampler. Also, we aim to learn more generalizable and scalable sampler so we use different parameterization and meta-learning strategy with Gong et al. Therefore, although we share some similar features with Gong et al., it is difficult to say our method is essentially very similar to Gong et al since they could not achieve our goals.

---

> ### Author Response · Authors · 2023-11-17
>
> > The motivation of the proposed method is weak. Since the main difference compared with Gong et al is the parameterization of the kinetic energy term, it is important to clearly motivate this choice. The authors did not mention at all in the paper why they choose to learn the kinetic energy term rather than the curl matrix and the diffusion matrix. The advantages of doing so are not discussed.
>
> Firstly, we would like to say that we noted the advantages of the design of kinetic energy over the design of diffusion and curl matrices in Appendix F along with ablation study. Also, we have added the motivation to learn the kinetic energy term in Appendix A.1. Parameterization of diffusion matrix $D_f$ and curl matrix $Q_f$ suggested in Gong et al., has two limitations to achieve our goal, efficient exploration of multi-modal large scale BNNs posterior distribution. First, learning $D_f$ and $Q_f$ to be dependent on $z$ harms scalability of algorithm since it introduces a new correction term $\Gamma_i(z)$ which forces to compute the gradient of $D_f$ and $Q_f$ with respect to $z$. This makes significant computational burden as the dimension of $z$ increases.
>
> Additionally, since $Q_f$ which governs the acceleration of $\theta$ is limited to operating as multipliers for the energy gradient and momentum, learning $D_f$ and $Q_f$ is not the best parameterization for learning sampler to effectively explore multi-modal distribution. In low energy regions where the norm of gradient and momentum are extremely small, it is difficult to make reasonable amount of update of $\theta$ for exploration by multiplying $Q_f$ to momentum and gradient. By contrast, In the update rule of L2E, two parameterized gradients $\alpha_\phi, \beta_\phi$ are added to the energy gradient and $\beta_\phi$ directly updates the $\theta$. This is more suitable for controlling the magnitude and direction of update to enhance exploration property in low energy regions.
>
> > Gong et al has proposed several meta objectives. Again, the authors did not discuss their meta objective with existing ones.
>
> To the best of our knowledge, Gong et al did not propose meta-objectives other than the meta-objective they actually used. However, we have added a discussion on the meta-objective proposed by Gong et al and other related papers in Appendix A.3. Meta-objective of Gong et al aims to reduce the KL divergence between the target density $\pi$ and the marginal distribution $q(\theta|D)$ at time $t$. However, $q(\theta|D)$ is intractable since it is an unknown density function, computing the gradient of $q(\theta|D)$ requires a gradient estimator. They used stein-gradient estimator which requires multiple independent chains, but this undermines the scalability of the algorithm. Also, this objective does not lead the learned sampler to explore multi-modal distribution. Figure 3 in Gong et al showed that learned sampler fastly converges to low energy region, but learned curl matrix restricts the amount of update in low energy region to prevent divergence which harms the exploration in low energy region.
>
>
> > In experiments, the authors did not compare with Gong et al, which is a very related method.
>
> In Appendix A.4 of revised version, we follow the experimental setup of Gong et al, and compare our method with Meta-SGMCMC (Gong et al). For experimental details and results, please refer to Appendix A.4. To summarize the experimental results, we have confirmed that L2E outperforms Meta-SGMCMC in various generalization experiments in terms of ACC and NLL. In CIFAR-10 experiment, despite L2E was not trained on the CIFAR-10 during the learning process, L2E significantly outperforms with a wide margin without being trained on CIFAR-10, indicating that our approach better generalizes to unseen datasets compared to Meta-SGMCMC.
>
>
> |Method|NT ACC|NT+AF ACC|NT+Data ACC|NT NLL/100|NT+AF NLL/100|NT+DATA NLL/100|
> |:---:|:---:|:---:|:---:|:---:|:---:|:---:|
> |Meta-SGMCMC|78.12±0.04|74.41±0.11|89.97±0.04|68.88±0.15|79.55±0.06|15.66±0.28|
> |L2E|**79.21±0.20**|**75.91±0.20**|**92.49±0.23**|**63.23±0.46**|**72.11±0.22**|**11.82±0.81**|
>
> |Method|NT ACC|NT+AF ACC|NT+Data ACC|NT NLL/100|NT+AF NLL/100|NT+DATA NLL/100|
> |:---:|:---:|:---:|:---:|:---:|:---:|:---:|
> |Meta-SGMCMC|98.36±0.02|97.72±0.02|98.62±0.02|640±6.25|875±3.19|230±3.23|
> |L2E|**98.39±0.05**|**98.07±0.08**|-|**558±3.19**|**679±6.65**|-|

---

> > ### Author Response · Authors · 2023-11-17
> >
> > > Although the experimental results of the proposed method are promising, it is not clear why the proposed meta-learning method can lead to these empirical improvements. What kind of learning updates did the meta-learning algorithm learn in the end? Why does the method show a better exploration-exploitation balance? how does it achieve that?
> >
> > We have added additional analysis of the behavior of L2E in Appendix B. Firstly, L2E shows good exploration-exploitation balance since L2E makes considerable amount of updates even in the local minima without increasing loss. This is different from other gradient-based MCMC methods where the update amount tends to decrease at minima. Considering that L2E collects samples from different modalities, we can say that our parameterized kinetic energy gradients learned to move around high density regions among different modalities.
> >
> > > The meta-learning tasks are classification on MNIST, Fashion-MNIST, EMNIST and MedMNIST and the downstream tasks are classification on Fashion-MNIST, CIFAR and Tiny ImageNet. Since Fashion-MNIST has appeared in meta-learning tasks, is it reasonable to use it as a test task? Are CIFAR and Tiny ImageNet similar to MNIST to be considered as test tasks?
> >
> > We added the Fashion-MNIST experiment to assess the performance of L2E on the in-distribution dataset included in meta-training tasks. However, since model architecture used for meta-training is different from that used in Fashion-MNIST experiment, we can still evaluate the generalization performance of L2E with respect to model architecture. Importantly, datasets such as CIFAR and Tiny-ImageNet are significantly different from the datasets in meta-training tasks, enhancing their value as test tasks. Evaluating L2E on completely different datasets and architectures can effectively showcase the transferability and generalization capabilities of L2E.
> >
> > > Can you explain T-SNE visualization of learning trajectories? Why does L2E’s trajectory look very different from DE and CSGMCMC?
> >
> > T-SNE visualization of learning trajectories is the method demonstrated by [1]. The authors of this paper aim to visualize the similarity in function space for each trajectory of DE. Therefore, we used T-SNE to visualize how L2E traverses the function space and collects samples. Figure 3 shows that the functions explored by L2E are far from the initial function, while simultaneously covering a wide range in function space within a single trajectory. The diversity of collected samples is supported by the results from KLD values in Table 1, cosine similarity heatmap in Figure 5 and linear interpolation plot in Figure 6.
> >
> > The difference in T-SNE plot between DE, CSGMCMC, and L2E is expected. DE starts from different random initializations for each trajectories so that there are several truncated trajectories of functions in T-SNE plot of DE. CSGMCMC makes artificial spikes of loss in the local minima through the cyclic learning rate schedule so the trajectories deviate from the highlighted region in Figure 3 with a large learning rate. In contrast, L2E does not initialize the chain or makes artificial spikes of loss while traversing through various regions of posterior distribution. In Figure 20, we show that L2E consistently keeps the loss and NLL as nearly constant while obtaining diverse samples from different modalities.
> >
> > [1] Fort, Stanislav, Huiyi Hu, and Balaji Lakshminarayanan. "Deep ensembles: A loss landscape perspective." arXiv preprint arXiv:1912.02757 (2019).

---

> > > ### Author Response · Authors · 2023-11-22
> > >
> > > Dear Reviewer LiEv,
> > >
> > > Thank you for your time in reviewing our work. Have we addressed your concerns raised in the initial review? The author discussion period will end tomorrow, so we would like to take the remaining time to ensure we have addressed any remaining concerns you may have. Please let us know if anything needs further clarification.
> > >
> > > Thank you, Authors

---

> > > > ### Comment · Reviewer_LiEv · 2023-11-23
> > > >
> > > > I appreciate the authors’ response, especially the new experimental results.
> > > >
> > > > **The parameterization of the kinetic energy**
> > > >
> > > > The benefits and motivation for parameterizing the kinetic energy are still unclear to me. I wonder if parameterizing the kinetic energy is theoretically sound. Since the kinetic energy depends on theta, my feeling is that the stationary distribution is no longer the target distribution. If so, what is the stationary distribution of the learned sampler? What are the requirements for the kinetic energy in order to ensure the Markov chain converges the target distribution? For parameterizing the diffusion and curl matrices, we know that the meta-learned sampler will still have the target distribution as the stationary distribution according to the complete recipe.
> > > >
> > > > **The meta-loss function**
> > > >
> > > > What is the reason that the proposed sampler can explore multi-modal distributions? It is not clear why meta-learning can help overcome the barrier between modes. The meta-loss does not seem to explicitly encourage this.
> > > >
> > > > Overall, I think the idea of a meta-learning sampler is very interesting and the experimental results seem promising. I would encourage the authors to continue revising the paper based on all reviews and submit this work to future venues.

---

### Official Review · Reviewer_CsK6 · 2023-10-31

**Soundness:** 3 good
**Presentation:** 3 good
**Contribution:** 3 good
**Rating:** 6
**Confidence:** 3

**Summary:**

In this paper, the authors present a general approach for SGMCMC using meta-learning. The meta-learning approach is explicitly trained to minimize the downstream log likelihood using Bayesian model averaging from the final model. This approach is motivated by the earlier successes of meta-learning approaches which show that rich feature representations can be learned and transferred to various tasks. The authors present experimental results looking at the convergence rate, the diversity of the samples, as well performance on downstream tasks of classification and uncertainty quantification.

**Strengths:**

- The approach is well motivated by the previous successes in meta-learning. Knowledge-sharing between different SGMCMC chains across different multi-modal distributions across similar tasks should be explored in detail.

- The authors have done a good job at experimentation overall. The empirical analysis spans understanding the diversity of MCMC chains as well as looking at the downstream tasks.

- I really like the idea of parameterizing the gradients of kinetic energy function - it seems intuitive and to the best of my knowledge I haven't seen prior work do that. This building the SGMCMC algorithm from first principles is commendable.

**Weaknesses:**

- While the experimental results are useful, I think the paper also needs an ablation study. We need to understand the impact of the parameterized gradients v/s transferability of the posterior information across the tasks.

- Understanding the compute requirement at train time is equally important. We don't know how much training time is required per step and in total and how it compares with other baselines that the authors have compared with. It'll also be useful to know the additional # of parameters added through the gradient parameterization.

- I think there will be some tradeoff between the number of samples we generate in the inner loop to compute $L$ and final convergence in terms of how quickly we converge and quality of the posterior samples. Running an ablation on the number of samples we generate in the inner loop would also be useful for practitioners who would be interested in using this approach.

**Questions:**

I do like this paper overall and I think it has shown some interesting results. I have listed my comments that I'd like the authors to address in the weaknesses section.

Also, it'll be useful if the author can provide some insights on how to speed up their algorithm and make it more efficient. For e.g., would warm starting with an SGD estimate in the meta-learning framework help?

---

> ### Author Response · Authors · 2023-11-17
>
> Thanks for your positive view supporting our method that enhances exploration of SGMCMC through learning kinetic energy terms. Our answer for your concern is as follows:
>
> > While the experimental results are useful, I think the paper also needs an ablation study. We need to understand the impact of the parameterized gradients v/s transferability of the posterior information across the tasks.
>
> We understand your question as asking “What is the common knowledge that parameterized gradients learn for the exploration of posterior distribution of different tasks”. If this is your question, we have added an analysis of the parameterized gradient during the evaluation stage in Appendix B for answering this. We track the norm of parameterized gradient, $||\beta||^2 = \frac{||\Delta\theta||}{\epsilon}^2$ we find that its magnitude is larger in local minima than in the early stages of training. This tendency is different from other gradient-based MCMC methods where the update amount decreases at minima. Additionally, we notice that L2E actively updates $\theta$ at minima while maintaining loss as nearly constant. This trend is consistently observed in both CIFAR-10 and CIFAR-100, implying that L2E learns some common knowledge of posterior information across tasks for efficient exploration in low loss regions.
>
> If this is not an appropriate answer to your question, could you please explain your question again in more detail?
>
> > Understanding the compute requirement at train time is equally important. We don't know how much training time is required per step and in total and how it compares with other baselines that the authors have compared with. It'll also be useful to know the additional # of parameters added through the gradient parameterization.
>
> For meta-training, we use approximately 10 hours for 2000 updates of meta-parameters on a single NVIDIA RTX A6000 GPU. It is important to note that we meta-train only once for all experiments in the paper since we aimed to meta-train the sampler which is transferable to various tasks. Therefore, it is unfair for our method to compare the wall clock time including meta-training time with other baselines.
>
> Our meta-learner consists of 2 layer MLP with 32 hidden nodes. This neural network takes a 9-dimensional input and output 2-dimensional vector so the additional number of parameters is 1442. This neural network is applied to each dimension of the inner-parameter independently. L2E takes about 1.2~1.5 times more wall clock time than CSGMCMC for one step update according to Table 18.
>
> > I think there will be some tradeoff between the number of samples we generate in the inner loop to compute L and final convergence in terms of how quickly we converge and quality of the posterior samples. Running an ablation on the number of samples we generate in the inner loop would also be useful for practitioners who would be interested in using this approach.
>
> We have added ablation experiments in Appendix C.3. for the number of samples in the inner loop. As you expected, we confirm that generating a smaller number of samples during inner-loop harms the convergence of the sampler in evaluation tasks. When the inner loop is too short, the sampler may not sufficiently learn information about the loss surface in low-loss regions during meta-training. Then, the sampler will fail to learn desirable properties such as exploration in minima or fast convergence.
>
> > I do like this paper overall and I think it has shown some interesting results. I have listed my comments that I'd like the authors to address in the weaknesses section. Also, it'll be useful if the author can provide some insights on how to speed up their algorithm and make it more efficient. For e.g., would warm starting with an SGD estimate in the meta-learning framework help?
>
> We can speed up L2E by reducing the thinning interval. As shown in Figure 19, while CSGMCMC shows performance drop as the thinning interval decreases, L2E can maintain a similar level of accuracy and diversity with shorter thinning interval. Specifically, L2E maintains performance even when reducing the thinning interval from 50 epochs to 10 epochs for both CIFAR-10 and CIFAR-100. L2E outperforms CSGMCMC's 5100 epochs performance with just 1100 epochs.
>
> Furthermore, L2E may benefit from using burn-in epochs with SGD, as it allows for warm starting with SGD and immediate sample collection using L2E to traverse high density regions between different modes. Additionally, since our approach shares some similarities with other SGMCMC in parameterization, we can apply techniques such as LR warm-up or LR scheduling. In other words, existing techniques for accelerating training can be applied to L2E so L2E has great potential in terms of efficiency and scalability. In conclusion, based on these insights and results, L2E has numerous ways to achieve speed up and potentially efficiency.

---

> > ### Comment · Reviewer_CsK6 · 2023-11-21
> > **Response to authors**
> >
> > I thank the authors for engaging in the discussion.
> >
> > - While talking about ablation study to understand the impact of parameterization v/s meta-learning, I meant what would be the performance impact if we didn't do parameterization and only used the meta-parameters as the starting point for sampling.
> > - After reading your responses, I am a little confused. What are the 9 inputs to the meta-network here?
> > - I guess 5100 epochs for cSGMCMC seems very high in general? Can you highlight what's the setup here? Also, the thinning interval of 50 or even 10 epochs seems pretty high. Especially, for cSGMCMC where you can potentially get away with very low thinning interval given that main benefits could come from running more cycles.
> >
> > I also read other reviews, and it seems like other reviewers also have concerns regarding the baseline. I also saw that the authors are not performing any data augmentation. For stronger baselines (such as deep ensembles), having data augmentations would be very important.

---

> > > ### Author Response · Authors · 2023-11-22
> > >
> > > Thanks for your response. Our answers for your questions are as follows.
> > >
> > > > While talking about ablation study to understand the impact of parameterization v/s meta-learning, I meant what would be the performance impact if we didn't do parameterization and only used the meta-parameters as the starting point for sampling.
> > >
> > > Our goal is completely different from other meta-learning methods that aim to learn good initial parameters which are adaptable for many tasks.  We aim to meta-learn the sampler that can explore highly multimodal posterior distribution in large-scale BNNs tasks. We believe that we cannot achieve our goal solely by meta-learning good initial parameters.
> > >
> > > > After reading your responses, I am a little confused. What are the 9 inputs to the meta-network here?
> > >
> > > For the meta-network, we feed the corresponding parameter and momentum values, the stochastic gradients of energy functions for that element, and the running average of the gradient at various time scales. Running average feature is expanded for multiple time scales in that we use multiple momentum-decay values for averaging. We use 0.1, 0.5, 0.9, 0.99, 0.999 and 0.9999 for momentum decay so that running average feature is expanded into 6-dimensions. Therefore, we have total 9-dimensional input features for each dimension of parameter and momentum.
> > >
> > > > I guess 5100 epochs for cSGMCMC seems very high in general? Can you highlight what's the setup here? Also, the thinning interval of 50 or even 10 epochs seems pretty high. Especially, for cSGMCMC where you can potentially get away with very low thinning interval given that main benefits could come from running more cycles.
> > >
> > > Basically, we use large thinning interval to collect diverse parameters from various parts of the posterior distribution. For cSGMCMC and L2E, we use 5100 epochs with 100 burn-in epochs and a thinning interval of 50 epochs to collect 100 parameters. For cSGMCMC, we also set cycle length to 50 epochs which is identical to the setting for CIFAR-10 and CIFAR-100 experiments of the original cSGMCMC paper [1]. Predictive performance of cSGMCMC decreases with a low cycle length since it requires a sufficient number of training steps for each cycle to converge to the low loss region. Also, due to the multi-modal and highly complex nature of the loss surface for large-scale BNNs posteriors, it is necessary to run many epochs to allow samplers to explore various parts of the posterior. We believe that collecting a diverse set of parameters and performing Bayesian Model Averaging (BMA) is important for BNNs. You can find other papers like [2]  that use more than 10000 epochs for collecting parameters in CIFAR-10 experiment.
> > >
> > > [1] Zhang, Ruqi, et al. "Cyclical stochastic gradient MCMC for Bayesian deep learning." arXiv preprint arXiv:1902.03932(2019).
> > >
> > > [2] Izmailov, Pavel, et al. "What are Bayesian neural network posteriors really like?." International conference on machine learning. PMLR, 2021.
> > >
> > >
> > > > I also read other reviews, and it seems like other reviewers also have concerns regarding the baseline. I also saw that the authors are not performing any data augmentation. For stronger baselines (such as deep ensembles), having data augmentations would be very important.
> > >
> > > We do not use data augmentation in our experiments since including data augmentation violates i.i.d assumption of the dataset. This can lead to model misspecification problem[1,2], which harms the performance of Bayesian methods. Also, [3] argued the incompatibility between data augmentation and Bayesian methods. Therefore, we do not use data augmentation to fairly compare the methods.
> > >
> > > [1] Wenzel, Florian, et al. "How good is the bayes posterior in deep neural networks really?."
> > >
> > > [2] Kapoor, Sanyam, et al. "On uncertainty, tempering, and data augmentation in bayesian classification." Advances in Neural Information Processing Systems 35 (2022): 18211-18225
> > >
> > > [3] Izmailov, Pavel, et al. "What are Bayesian neural network posteriors really like?." International conference on machine learning. PMLR, 2021.

---

> > > > ### Comment · Reviewer_CsK6 · 2023-11-22
> > > > **Response to authors**
> > > >
> > > > Thanks for your comments.
> > > >
> > > > > Our goal is completely different from other meta-learning methods that aim to learn good initial parameters which are adaptable for many tasks. We aim to meta-learn the sampler that can explore highly multimodal posterior distribution in large-scale BNNs tasks. We believe that we cannot achieve our goal solely by meta-learning good initial parameters.
> > > >
> > > > But this would still be a good baseline, I believe. And this would help showcase why parameterized gradients are needed. I see that you also did another experiment looking at diversity of datasets, which is also helpful.
> > > >
> > > > > Basically, we use large thinning interval to collect diverse parameters from various parts of the posterior distribution. For cSGMCMC and L2E, we use 5100 epochs with 100 burn-in epochs and a thinning interval of 50 epochs to collect 100 parameters. For cSGMCMC, we also set cycle length to 50 epochs....
> > > >
> > > > So L2E and cSGMCMC are run for identical number of epochs? If so, why is the training time different, exactly? Also, what I was alluding to previously was that you can reduce the thinning interval for cSGMCMC and I also believe that after a few # of cycles, the performance gain from additional parameter samples would be negligible. So we may not need very high # of parameter samples in the first place.
> > > >
> > > > > We do not use data augmentation in our experiments since including data augmentation violates i.i.d assumption of the dataset.
> > > >
> > > > I get that it's not a principled thing to do - however, as one other reviewer has pointed out, it significantly impacts the performance. One way to get around this is to say that after augmentation, you're generating samples from a different distribution (as long as the support for original samples under that distribution doesn't go down to zero). Agreed that this can still invite arguments against biases, but this would be one way to get around that assertion.

---

### Author Response · Authors · 2023-11-17

# Overall Response

We appreciate the efforts for reviewing our paper. We have revised the original paper reflecting the points raised by the reviewers. We mark the revision parts with sky blue color. Please check the revised version of our paper. Thank you.

We would like to answer some common questions in the overall response.

- **Limitation of previous work (Gong et al.(2018)) for meta-learning SGMCMC.**

Although Gong et al. firstly proposed a meta-learning based approach for SGMCMC, this algorithm shows several limitations to achieve our goal, learning the generalizable sampler which efficiently explores the multi-modality of large scale BNNs posterior. Firstly, Gong et al. trained the sampler only on a single task, which significantly reduces the generalizability of their method.  Secondly, their algorithm shows weakness in terms of scalability, since parameterization of diffusion and curl matrix incurs additional computation cost for correction terms. Finally, their meta-objective did not encourage exploration in high density regions. We have added comprehensive comparison with Gong et al., in Appendix A.

- **Why do we choose to parameterize the kinetic energy?**

Parameterizing the kinetic energy has two advantages over parameterizing diffusion matrix $D_f$ and curl matrix $Q_f$. Firstly, it is more scalable since it can avoid the additional computation of the correction term. Correction term, $\Gamma_i(z)$,  includes gradient computation which scales linearly with the dimension of $z$. Secondly, since diffusion and curl matrices mainly operate as multipliers for gradient and momentum, it is difficult to make reasonable amount of update of $\theta$ in local minima where gradient and momentum is extremely small. In contrast, when modeling kinetic energy gradients,  two parameterized gradients $\alpha_\phi,\beta_\phi$ are added to the energy gradient and $\beta_\phi$ directly updates the $\theta$. This is more suitable for controlling the magnitude and direction of update to enhance exploration property in low energy regions.

- **Training cost for meta learning procedure**

It is important to note that we meta-train only once for all experiments in the paper so we do not need additional cost to train parameterized gradients in L2E at the evaluation stage. Therefore, we argue that it is inappropriate to include the meta-training cost when comparing running time with other baselines. Also, our meta-training takes about 10 hours for meta-training on a single NVIDIA A6000 GPU, which is relatively small compared to other recent works conducting multi-task meta-training like [1].


- **Our contribution**

Main contribution of our work is that we propose the parameterization and meta-learning strategy for training generalizable sampler that effectively captures multi-modality of large-scale Bayesian Neural Networks(BNNs) posterior across various unseen tasks through meta-training only once. Since existing works have limitations such as lack of scalability, ineffective exploration of multi-modality and limited generalization to unseen tasks, we think our contribution is clear and meaningful enough.

[1] Metz, Luke, et al. "Meta-learning update rules for unsupervised representation learning." arXiv preprint arXiv:1804.00222 (2018).

# Errata
- Revise introduction section for discussing Gong et al.
- Add new appendix section A for comparing L2E with Meta-SGMCMC
- Add new appendix section B for analysis of exploration property of L2E
- Add new appendix section C for 1-D synthetic regression experiment, data augmentation experiment and additional ablation studies
- Add new appendix section D for related works

---

### Meta-Review · Area_Chair_GyBF · 2023-12-07

**Metareview:**

The reviewers agree that there are interesting points in this paper, but feel that it needs more work to set itself apart from existing works, such as Gong et al. (2018). In particular, a head-to-head comparison of the proposed method and the existing method following the experiment setting of Table 1 would be quite useful. It would be appealing if the proposed method has a significant improvement over the existing ones.

**Justification For Why Not Higher Score:**

If in the experiments, the proposed method can be compared to the existing work and have a large margin of improvement, I would vote for the acceptance of this paper.

**Justification For Why Not Lower Score:**

n/a

---

### Decision · Program_Chairs · 2024-01-16

Reject